# SMRT-Cappable-seq reveals complex operon variants in bacteria

Bo Yan[1], Matthew Boitano[2], Tyson A. Clark [2] & Laurence Ettwiller [1]

Current methods for genome-wide analysis of gene expression require fragmentation of original transcripts into small fragments for short-read sequencing. In bacteria, the resulting fragmented information hides operon complexity. Additionally, in vivo processing of transcripts confounds the accurate identification of the 5′ and 3′ ends of operons. Here we develop a methodology called SMRT-Cappable-seq that combines the isolation of unfragmented primary transcripts with single-molecule long read sequencing. Applied to *E. coli*, this technology results in an accurate definition of the transcriptome with 34% of known operons from RegulonDB being extended by at least one gene. Furthermore, 40% of transcription termination sites have read-through that alters the gene content of the operons. As a result, most of the bacterial genes are present in multiple operon variants reminiscent of eukaryotic splicing. By providing such granularity in the operon structure, this study represents an important resource for the study of prokaryotic gene network and regulation.

[1] New England Biolabs Inc., 240 County Road, Ipswich, MA 01938, USA. [2] PacBio, 1305 O'Brien Drive, Menlo Park, CA 94025, USA. Correspondence and requests for materials should be addressed to L.E. (email: ettwiller@neb.com)

The model of operon structure proposed by Jacob and Monod invoked a group of contiguous genes regulated by a common promoter in bacteria[1]. The resulting polycistronic transcripts can be several kilobases (kb) long and encode several proteins usually associated with the same function or metabolic pathway.

With decades of research on bacterial transcriptional regulation, this operon model has been found to be complex with a rich landscape of mechanisms to control expression. RNA-seq and microarrays have been instrumental in understanding many of these mechanisms. But, while these technologies are great in interrogating genome-wide expression profiles in windows of hundreds of bases, they do not provide information on the larger transcriptional context (TC) typically found with bacterial operons. This shortcoming in current technologies for transcriptome analysis has impaired our ability to delineate transcript starts and ends that are typically several kb apart.

To access these transcriptional landmarks that define transcription start sites (TSS) and transcription termination sites (TTS), strategies have been developed to specifically identify TSS[2–5] and TTS[6] at base resolution. Nonetheless as for RNA-seq, these strategies rely on the necessary fragmentation of transcripts for short-read sequencing. Consequently, the larger context by which genes are expressed and the phasing between starts and the ends of primary transcripts are lost.

In this study we have developed a strategy called SMRT-Cappable-seq that combines the isolation of the full-length prokaryotic primary transcriptome with PacBio SMRT (Single Molecule, Real-Time) sequencing. Sequencing E.coli transcriptome using SMRT-Cappable-seq reveals complex operon structures originated notably from the widespread existence of read-through at termination sites. Such read-through can be modulated according to growth conditions, highlighting a possible regulatory mechanism for gene expression.

## Results

### Sequencing full-length transcripts using SMRT-Cappable-seq.
To ensure the sequencing of full-length primary transcripts, we capture the triphosphorylated 5′ ends of transcripts matching the TSS. For this, we adapted Cappable-seq technology previously used to identify TSS[2] for the isolation of long transcripts (Fig. 1a). The principle of the Cappable technology is based on specific desthio-biotinylation of the 5′ triphosphate characteristic of the first nucleotide incorporated by the RNA polymerases[2]. The desthio-biotinylated RNA is then captured on streptavidin beads and subsequently released from the beads after several washing steps to remove processed RNA.

The capturing of the 5′ triphosphate is expected to markedly enrich for primary transcripts that have also retained their original 3′ ends. Indeed, since the first step of most in vivo RNA degradation pathways is thought to consist of the removal of the 5′ triphosphate, the capturing of triphosphorylated RNA removes degraded and/or processed transcripts on the 3′ end, particularly ends generated from RNase E processing[7]. Nonetheless additional nucleases that do not require the removal of the 5′ triphosphate have been shown to exist[8] and thus, some remaining 3′ends in our data set may be derived from processing.

To sequence the most 3′end of the captured transcripts, a polyA tail is added and cDNA is synthesized via reverse transcription (RT) using an anchored polyT primer (Fig. 1a and Supplementary Fig. 1). After RT reaction, a polyG is added to the 3′end of the cDNA using terminal transferase. Second-strand synthesis is performed using a polyC primer. Finally, the unfragmented cDNA is size selected for large fragments (>1 kb), amplified and sequenced using PacBio long read sequencing

technology resulting in the identification of full-length transcripts at base resolution. Importantly, long read sequencing provides the phasing of both ends of single transcripts, overcoming the inabilities of short reads to obtain long-range continuity (Fig. 1b). Thus, SMRT-Cappable-seq provides a powerful approach for directly identify entire operon at molecule resolution in bacteria.

We applied SMRT-Cappable-seq to the un-fragmented total RNA from E. coli grown in minimal (M9) and Rich medium to compare how growth conditions affect the transcriptome. We combined both data sets to obtain a comprehensive overview of the transcriptional landscape of long transcripts (>1 kb) in E. coli. The combined data set consists of a total of half a million sequencing reads with an average length of 2000 bp. More than 99% of reads are mapped to the E. coli genome and only 0.2% of the reads show chimeric structures and were discarded (Methods). With a coverage of 90.3% of the genome and 81% of the known genes covered entirely by at least one read, SMRT-Cappable-seq provides a comprehensive view of the bacteria full-length transcriptome. Despite size selection of the SMRT-Cappable-seq library favoring long fragments, we found a decent correlation (Spearman's rank correlation 0.798, p value < 2.2e-16) between gene expression derived from SMRT-Cappable-seq and published Illumina RNA-seq[9] (Fig. 1c and Supplementary Note 1) indicating that SMRT-Cappable-seq is suitable for quantitative measurement of transcript levels. In order to evaluate the reproducibility of the method, we have also performed a separate technical replicate (Methods) and obtained a very good correlation between replicates (Supplementary Note 2) in terms of gene expression (Supplementary Fig. 2a), TSS and TTS identification and quantification (Supplementary Fig. 2b, c).

To assess the depletion of processed transcripts, which is an important step in the SMRT-Cappable-seq procedure, we analysed the fraction of reads mapped to ribosomal RNA (rRNA). In bacteria, rRNAs are formed by the processing of the single pre-rRNA transcript to form the mature 5S, 16S, and 23S rRNAs[10]. Because processed rRNA accounts for the vast majority of the RNA in the cell[10], differential levels of processed rRNA in the control versus SMRT-Cappable-seq libraries is a good indicator of the specificity of SMRT-Cappable-seq for primary transcripts. We found that the number of reads from processed rRNA drops from 71% in the control library to 4% in the SMRT-Cappable-seq library (Fig. 1d and Supplementary Note 3). Furthermore, amongst the reads mapped to rRNA, only 0.4% of rRNA reads in the control library correspond to primary rRNA transcripts (based on the 5′end location) versus up to 53% in the SMRT-Cappable-seq library (Supplementary Note 3). Accordingly, the fraction of mappable reads matching known 5′ or 3′ processing sites is only 2% in the SMRT-Cappable-seq libraries, sharply contrasting with 33% in the control library (Supplementary Notes 3, 4).

These results were validated by quantitative PCR (qPCR) experiments designed to measure the absolute recovery of primary and processed transcripts, before and after streptavidin enrichment. By measuring the absolute amount of RNA molecules, qPCR provides a more accurate estimation of the enrichment for primary transcripts compared to sequencing. qPCR results reveal that an average of 24% of primary transcripts are recovered after streptavidin enrichment compared to only an average of 0.02% recovery rate for processed rRNAs (Supplementary Fig. 2d). Based on these qPCR results, we conclude that SMRT-Cappable-seq has a 1200-fold greater recovery of primary transcripts compared to processed RNAs. Taken together, our results demonstrate that SMRT-Cappable-seq enriches primary transcripts and avoids sequencing of processed transcripts such as rRNA, tRNA, and others.

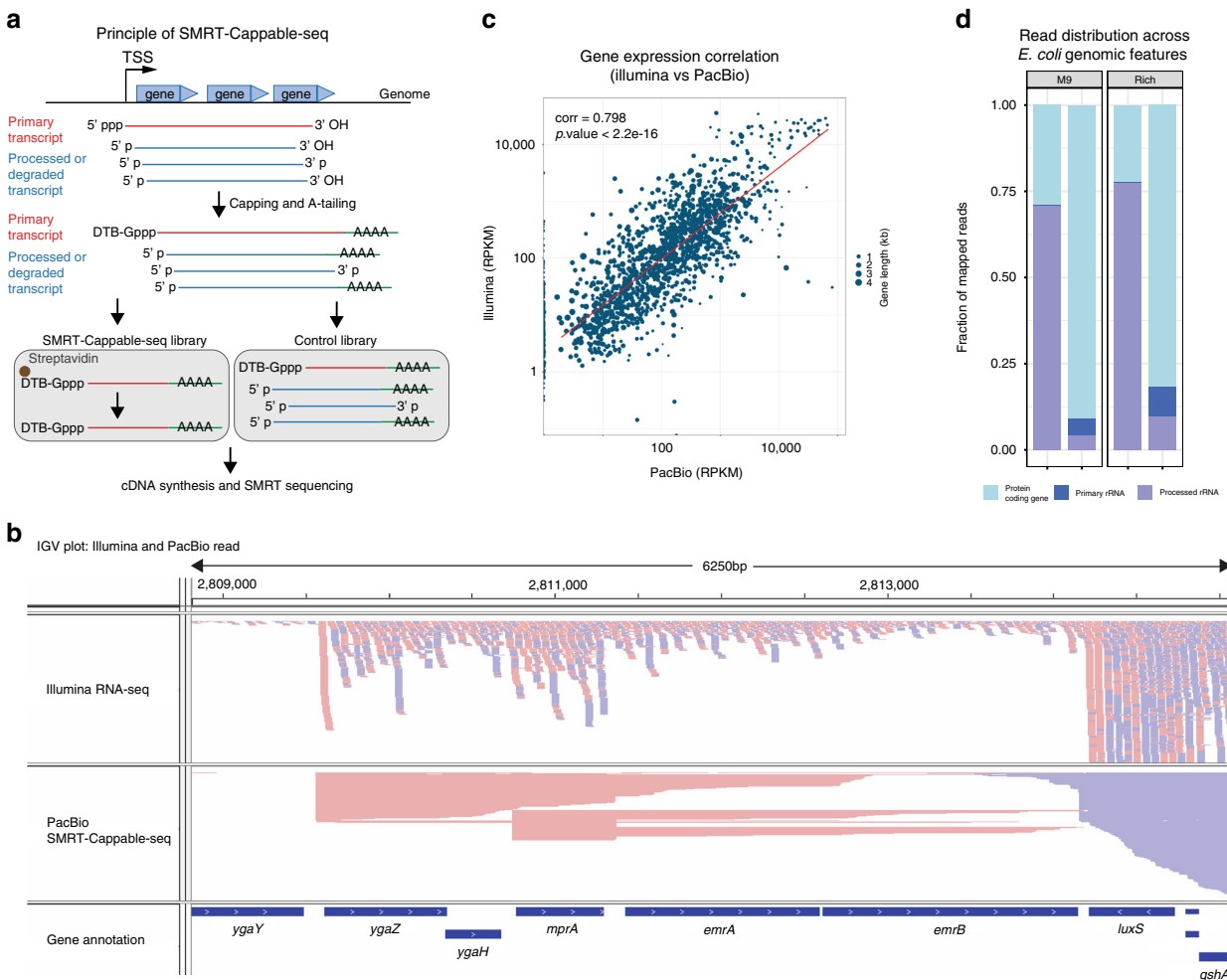

**Fig. 1** SMRT-Cappable-seq identifies full-length transcripts in bacteria. **a** Schema of the SMRT-Cappable-seq methodology. 5′ triphosphorylated transcripts are capped with a desthio-biotinylated (DTB) cap analog and bound to the streptavidin beads to specifically capture primary transcripts starting at TSS. The polyadenylation step (A-tailing) ensures the priming of the anchored poly dT primer for cDNA synthesis at the most 3′end of the transcript. **b** Integrative Genomics Viewer (IGV) representation of the mapping of SMRT-Cappable-seq reads (top) compared to Illumina RNA-seq reads (bottom) in the *mprA* locus. Forward oriented reads are labeled in pink, reverse oriented reads are labeled in blue. **c** Comparison between gene expression level in Read counts Per Kilobase of transcript, per Million mapped reads (RPKM) for Illumina RNA-seq and SMRT-Cappable-seq. The Spearman's rank correlation is 0.798 (*p* value < 2.2e-16). Point size denotes the size of the gene (in kb). **d** Fraction of reads mapped to protein coding genes (light blue), primary rRNA (dark blue), and processed rRNA (purple) for both M9 and Rich growth conditions (left panel and right panel, respectively). Reads mapped to primary rRNA are defined as reads which start at a known TSS of a primary rRNA transcript[2]. Processed rRNAs correspond to reads mapped to the rRNA genes but do not start at these TSSs

**SMRT-Cappable-seq identifies transcription landmarks**. Consistent with the enrichment for primary transcripts, we found that 93% of mapped SMRT-Cappable-seq reads start within 5 bp of a previously annotated TSS[2] compared to only 15% in the control library (Supplementary Note 5). We define TSS at single base resolution as genomic positions containing a significant accumulation of 5′end reads. If several TSSs were found within a 5 bp window, only the TSS position with the highest accumulation of read was retained (Methods). Libraries derived from M9 and Rich growth conditions have a total of 2186 and 1902 TSSs, respectively. Comparison between growth conditions reveals that 1350 TSSs are common and the TSS usage is globally correlated (corr = 0.3, *p* value < 2.2e-16). The preferred nucleotide for TSS is either adenosine or guanosine consistent with the literature[11], and nucleotide bias at −10 and −35 bp reveals the standard profile of bacterial promoters (Supplementary Fig. 3a). Collectively, these results confirm the 5′end completeness of the majority of transcripts in SMRT-Cappable-seq.

Similar to the TSS, termination sites were defined at single base resolution as genomic positions containing a significant accumulation of 3′end reads. Contrasting with the 5′end for which the labeling of the triphosphate ensures the direct identification of TSS, the capturing of the termination site is indirect and therefore requires more stringent filtering criteria (Methods). Consequently, we observed 347 highly confident termination sites in M9 condition, among which 74 and 1 have been previously identified as Rho-independent and Rho-dependent termination sites, respectively (Supplementary Data 1). Very similar results were obtained for the Rich condition (Supplementary Data 1 and Supplementary Notes 6, 7). To estimate the fraction of true TTS, we compared our defined termination sites with the RNase III cleavage sites, a post-transcriptional step that generates processed 3′ ends[12], and found only 7 SMRT-Cappable-seq termination sites overlapping with Rnase III cleavage sites (Supplementary Note 4). While RNase E also generates processed 3′ ends, the 5′ triphosphate transcripts targeted by SMRT-Cappable-seq are known to be resistant to RNase E processing[13]. Thus, SMRT-Cappable-seq transcripts ends should

not be resulting from RNase E cleavage. Accordingly, 95% of the SMRT-Cappable-seq termination sites have a predicted stem-loop structure characteristic of termination sites (See below and Supplementary Fig. 3b). Furthermore, analysis of the sequence upstream of the termination sites identifies two GC rich regions at −24 to −20 bp and −10 to −7 bp followed by a run of uracil residues characteristic of the Rho-independent TTS (Supplementary Fig. 3c)[14]. Taken together, these results suggest that most of the termination sites identified by SMRT-Cappable-seq are genuine TTS.

The majority (86%) of termination sites defined by SMRT-Cappable-seq are located in intergenic regions and almost all are located downstream of one or multiple gene(s), positioning these termination sites at the 3′UTR ends of mature transcripts (See the example shown in Fig. 2a). The relatively low abundance of premature termination sites in SMRT-Cappable-seq libraries is contrasting with the large number of known regulatory sites controlling transcript elongation in the 5′UTR of genes[14,15]. This

inconsistency can be explained by the size selection step during library preparation depleting prematurely terminated transcripts that are typically very short, below the selected 1 kb threshold. We nonetheless identify premature transcripts derived from well-known examples of termination sites in the 5′UTR region of the operon such as the Leu operon (Supplementary Fig. 4). Summing up, our results demonstrate the ability of SMRT-Cappable-seq to comprehensively identify full-length primary transcripts at base resolution. By essentially targeting primary transcripts, SMRT-Cappable-seq is ideally suited to analyse transcriptional output and operon structures in bacteria.

**34% of operons in RegulonDB are extended**. To take full advantage of the unique abilities of SMRT-Cappable-seq to obtain long-range continuity of transcripts, we simultaneously analysed the positions of 5′ and 3′ ends of reads representing unique transcripts in *E. coli*. Surprisingly, we observed that a large

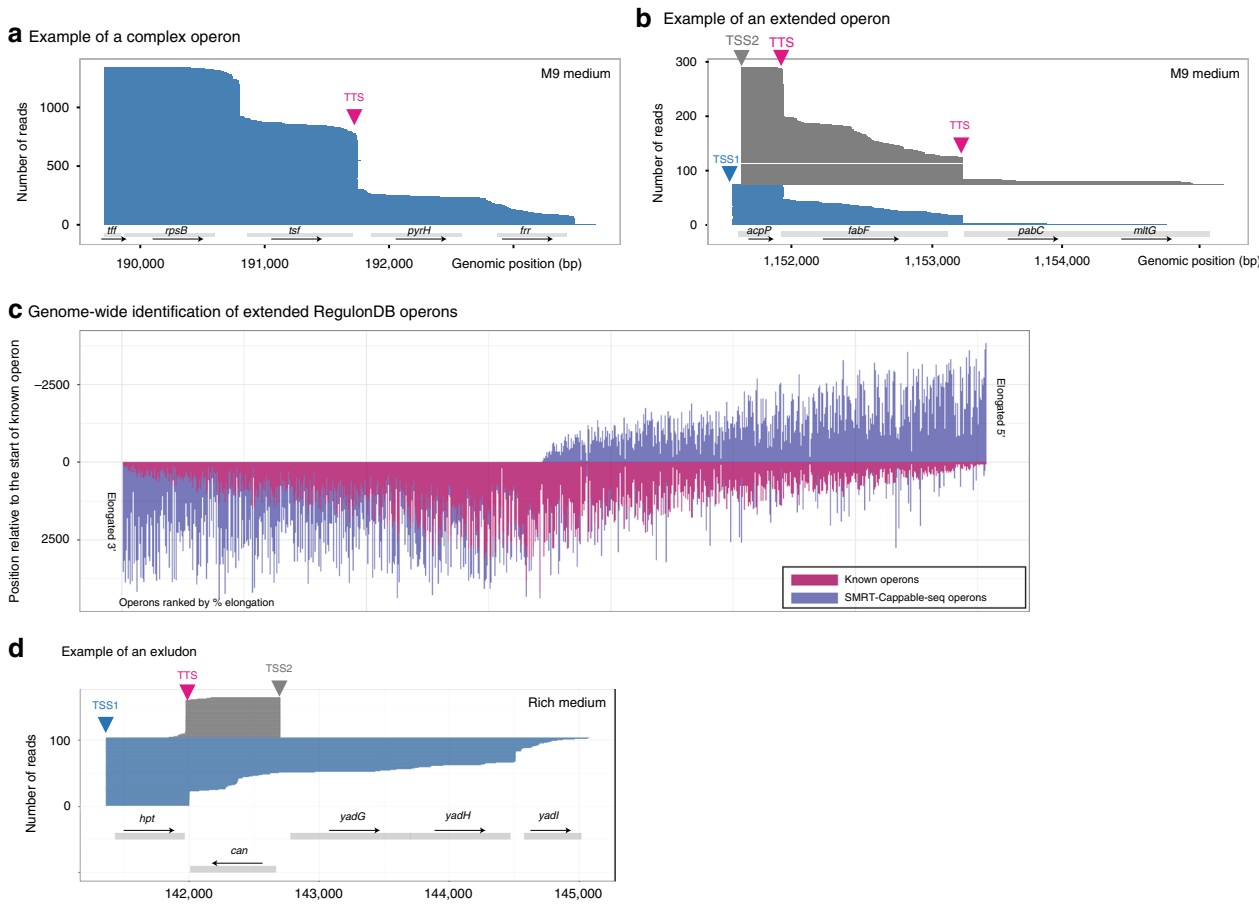

**Fig. 2** Operons in *E. coli*. **a** An example of the *tff-rpsB-tsf* operon in RegulonDB elongated at its 3′end with two additional genes: *pyrH* and *frr*. The *x*-axis represents the position (in bp) on the reference genome (NC_000913.3) and the *y*-axis represents individual mapped reads ordered by read size in ascending order. For clarity only reads from one TSS (position at 189712) are shown. Red arrow indicates the previously described Rho-independent terminator. *tff* encodes a putative small RNA. The product of *pyrH* is involved in nucleotide biosynthesis, while the products of rpsB, *tsf*, and *frr* are involved in translation. **b** An example of extending the predicted operon *acpP-fabF*[18] with two additional genes: *pabC* and *mltG*. **c** Genome-wide distribution of elongated operons compared to RegulonDB with additional gene(s). Elongated operons are defined by at least one SMRT-Cappable-seq read that covers the entire known operon and extends it to include at least one additional fully covered gene. Only the longest known operons were used, and sub-operons fully included in the longest operons were excluded from this analysis. Each line corresponds to the size of the operons (in bp) with the previously annotated operons in pink and the extended operons in blue. Positions are relative to the 5′end of the annotated operon with 0 being the TSS of the annotated operons. In total there are 883 RegulonDB operons extended by SMRT-Cappable-seq from either 5′ or 3′ end or both. **d** An example of an exludon. The transcripts encoding the *can* gene in the sense direction (gray) overlap with the transcripts (blue) containing the entire *can* gene in the antisense direction. Both transcripts share a bidirectional terminator (red arrow) with differential read-through where the majority (93%) of antisense transcript coding for the *can* gene terminates while the majority (85%) of the transcripts coding for *hpt* has read-through

fraction of reads extend further from annotated termination sites in RegulonDB and often include additional genes.

To systematically evaluate the fraction of extended operons containing additional genes, we compared transcripts defined from the combined SMRT-Cappable-seq libraries with known operons from RegulonDB[16]. If at least one SMRT-Cappable-seq read fully covers and extends a RegulonDB operon by one or multiple gene(s) on the 3′, 5′, or both ends, this RegulonDB operon is defined as extended (Methods). Using this definition, we found that 34% (883 operons) of annotated operons in RegulonDB are extended (Fig. 2a, b, c). With new polycistronic operons containing additional genes, SMRT-Cappable-seq represents a powerful technique to identify full-length transcripts and transcriptional boundaries. Because genes implicated in same metabolic pathways are often found on the same operon[1], SMRT-Cappable-seq also provides a powerful strategy to predict with experimental evidence for co-expression, metabolic pathways, and gene function in bacteria.

While *E. coli* is one of the best annotated prokaryotic species, it also has a total of 924 genes of yet unknown or predicted function (based on RegulonDB annotation)[16]. Based on our data set, 110 of these unknown genes are now found together with genes with known function in the extended operons. For example, two previously unknown genes *yedJ* and *yedR* are found on the same operon with *dcm* and *vsr*, which encode DNA-cytosine methyltransferase and DNA mismatch endonuclease, respectively. Accordingly, we predict the *yedJ* and *yedR* genes to be involved in processing DNA modification or DNA repair.

Finally, we also found many operons (165 in M9 and 146 in Rich) which contain entire genes in both the sense and the antisense direction (see Fig. 2d for example of such operon). This result confirms the widespread presence in *E. coli* of long antisense RNAs (asRNAs) described for *Listeria*[17].

**SMRT-Cappable-seq reveals a rich architecture of variants**. Extended operons correspond to only a subset of differentially annotated operon in SMRT-Cappable-seq data sets. With the ability to link the 5′ to the 3′ end of transcripts, SMRT-Cappable-seq unravels a number of smaller operons within operons leading to transcripts with novel combination of genes.

By analogy with the variety of eukaryotic transcripts resulting from splicing, we describe such modular usage of operons as operon variants. To avoid overestimating the number of operons, we use a conservative definition of operons for which a defined TSS and a defined TTS can be found. If no TTS can be found, the longest transcript is used to describe operons (Methods).

Thereby, we defined 2347 operons that encode unique combination of genes, amongst which 840 (36%) are not annotated in the RegulonDB database (Supplementary Data 2, Data 3 and Supplementary Note 8). We also compared the SMRT-Cappable-seq operons with operons predicted in a previous study[18], and we found around 40% of our operons were not predicted (see Fig. 2b for example). SMRT-Cappable-seq operons have an average of 2.1 genes per operon compared to 1.4 in RegulonDB suggesting that polycistronic operons are more prevalent than previously thought: around 30% of the defined operons start within another operon (see Supplementary Fig. 5 for examples).

To assess the extent of operon variants in *E. coli*, we define transcriptional context (TC) as the unique combination of genes within an operon, and investigated for each annotated gene, the number of unique TC the gene is found in. We found that around 50% of genes analysed are in more than one TC with some genes being found in, as many as 15 TCs (Fig. 3a). On genome-wide scale, SMRT-Cappable-seq identifies 2370 more

genes in multiple TCs than those annotated in RegulonDB. The Ribosome-recycling factor *frr* gene exemplifies the extent of TC modularity with eight different TCs (Fig. 3b), each of them having a different gene configuration. In contrast, RegulonDB and previous literature describe only the transcript containing the *frr* gene alone[19]. These results demonstrate that genes within multiple TCs are more common in bacteria than previously thought, highlighting a level of gene control that has not yet been thoroughly assessed.

Interestingly, TC can be condition dependent. For genes that are expressed in both M9 and Rich media, 35% display differences in TC. Those genes tend to be found in a greater variety of TC in Rich compared to M9 medium (Fig. 3c). For example, the *rplB* gene coding for the 50S ribosomal subunit protein L2 is transcribed in 13 different TCs when the cells are grown in Rich medium compared to only 3 in M9 medium (Fig. 3d). Genes with more TCs in Rich medium (Supplementary Data 4) are significantly enriched in biological processes involved in translation (GO analysis, 12-fold enrichment, $p$ value $= 1.62e\text{-}09$ with Bonferroni correction for multiple testing). This result suggests that the overall operon complexity can depend on the growth condition. Nonetheless further work needs to be done to rule out the possible confounding effect of changes in gene expression.

**Operon variants result from extensive read-through at TTS**. Next, we investigated the possible mechanisms underlying the existence of the large number of operon variants found by SMRT-Cappable-seq. We found that many of the sub-operons and extended operons that we identify are the consequence of extensive read-through at TTS. The concept of read-through is not new and has been demonstrated for some operons[20–22], reporter constructs[23] and described for convergent operons[18]. Nevertheless, the extent of read-through across prokaryotic transcriptome or the link between TTS read-through and operon variants has never been explored.

SMRT-Cappable-seq highlights the pervasiveness of the in vivo read-through of the entire transcriptome with 75% of TTSs having at least 5% of read-through transcripts (Fig. 4a, Supplementary Data 2 and Supplementary Note 7). TTSs overlapping with experimentally validated TTSs show essentially similar behavior with 70% of known terminators having read-through (Fig. 4a). The degree of read-through largely depends on the TTS, in some cases as high as 50% of the transcripts extending across sites (Fig. 4a and Fig. 2a). Interestingly, the extension in the majority of the cases is longer than 100 bp on average (Fig. 4b, c), and around 40% of termination sites have read-through transcripts that include additional gene(s), which indicates that the downstream extension is functional (Fig. 4c). As a result, display of transcripts sharing the same TSS collectively forms a staircase pattern originating at the 3′ end (Supplementary Fig. 6).

Comparison between growth conditions reveals that, in a notable number of cases, the degree of read-through is condition dependent (Fig. 5a), consistent with TTS serving as a major modulator of gene regulation. For example, the *prs-dauA* operon containing a TTS between the *prs* and *dauA* gene shows 91% read-through in Rich medium as opposed to only 46% in M9 condition (Fig. 5b and Supplementary Fig. 7a for more examples). This change in the degree of read-through results in the apparent up-regulation of the *dauA* gene in Rich condition despite similar expression levels at the TSS. The *dauA* gene encodes for the C4 dicarboxylic acid transporter, which mediates succinate transportation. Previous studies have shown that the acetate accumulation rate is faster in Rich compared to M9 medium[24], which leads to an acidic pH. As a result, succinate has to be excreted from the cell, which requires the expression of the *dauA*

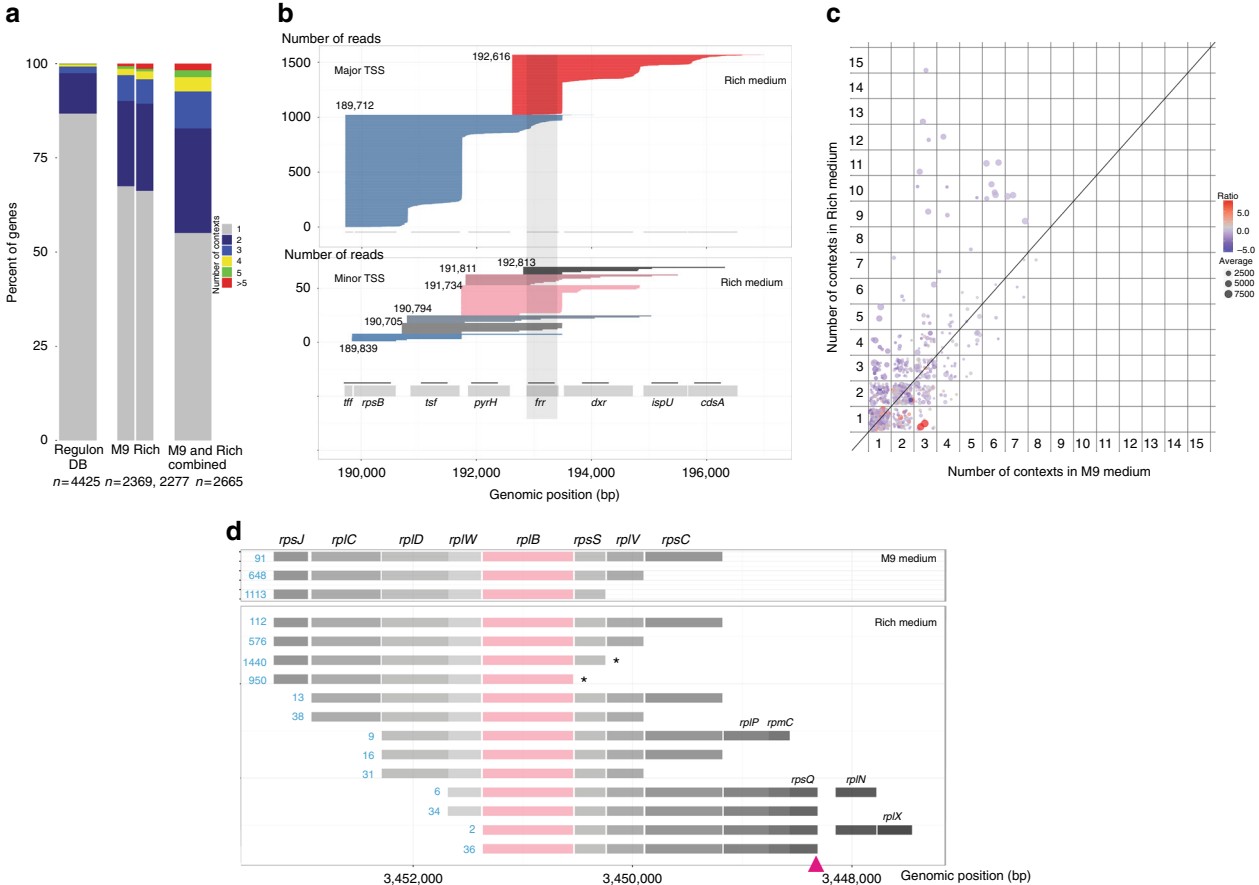

**Fig. 3** Transcriptional context. **a** Distribution of genes according to the number of transcriptional contexts from RegulonDB (left Bar chart), individual SMRT-Cappable-seq (Center Bar charts), and combined SMRT-Cappable-seq data set (Right Bar chart). Only genes having at least one context are considered (n = the number of genes). The number of transcriptional contexts for a given gene X is the number of unique combinations of genes co-transcribed with gene X. **b** Individual reads (Rich condition) mapping across the *frr* gene ordered by TSS location and decreased read size. Read colors represent TSS positions. Reads derived from major and minor TSS are divided into two panels offering different scale for clarity. The *frr* gene is transcribed from eight different gene contexts when the cells are grown in Rich medium. These contexts are altering the gene content of the transcripts encoding the frr protein. The eight possible combinations are: *rpsB-tsf-pyrH-frr*, *tsf-pyrH-frr-dxr*, *tsf-pyrH-frr*, *pyrH-frr-dxr*, *pyrH-frr*, *frr-dxr-ispU-cdsA*, *frr-dxr-ispU*, and *frr* alone. RegulonDB only annotates *frr* alone. **c** Number of contexts in M9 medium (x-axis) compared to the number of contexts in Rich medium for each selected genes. The point size represents the average expression level between the two growth conditions while the color represents the ratio between the expression level in Rich medium compared to M9 medium. **d** Example of the *rplB* gene coding for the 50S ribosomal subunit protein L2 showing a change in the number of transcriptional context from 13 contexts (Rich medium) to only three contexts (M9 medium). * indicates the SMRT-Cappable-seq defined TTSs downstream of *rpsS* and *rplB*, and red arrow indicates the previously known TTS downstream of *rpsQ*. The number of reads supporting each context is shown in blue

gene[25]. Therefore, the control of read-through of the TTS is a possible mechanism to enable the bacteria to adjust their enzyme levels quickly to the fast changing environment.

Furthermore, we found for a handful of cases that the degree of read-through of a given terminator can vary according to the TSS, a phenomena that we term "TSS-dependent read-through", because the level of TTS read-through is correlated with the location of the transcript TSS (Fig. 5c). Accordingly, we hypothesize that the overall structure difference between transcripts initiated at different TSS maybe a mechanism for such coordination.

Next we examined whether this pervasive read-through across TTS is the consequence of "weak" terminators. For this, we predicted the termination loop structure, and found no significant correlation (Pearson correlation 0.022, p value = 0.6487) between the free energy of the RNA secondary structure and the degree of TTS read-through (Supplementary Fig. 3b). Furthermore, a similar motif profile containing the classic loop structure was observed for TTS data sets with both low and high read-through

(Supplementary Fig. 3b and 3c). The polyU characteristic of the region of the termination sites is present for both high and low read-through TTS, however, and consistent with previous findings, we found that the polyU tract is better defined in the low read-through TTS data set (Supplementary Fig. 3c)[26]. Collectively, these results indicate that the structure of the termination loop tends to be well-conserved with a better defined polyU track for the stronger TTS.

Next, we examined whether regions flanking the loop can be another determinant of the read-through level. Previous studies have described riboswitches leading to two mutually exclusive RNA structures, one of which forms a transcriptional terminator and results in transcription termination, and the other forms an antiterminator that allows read-through across the terminator[14,27]. While riboswitches have been extensively studied in the past, they have primarily been identified in the 5′ UTRs of operons to modulate the commitment of the RNA polymerase to either fully transcribe or prematurely abort transcripts[14]. We analysed the TTS found by SMRT-Cappable-seq using a

published algorithm for the prediction of regulatory premature termination such as attenuator[28].

We found that 13.5 % of the TTSs are preceded by predicted attenuator structures (Fig. 5d). That is a 10 fold enrichment of attenuator structures at SMRT-Cappable-seq TTS compared to randomly selected sequences, suggesting that the read-through may be modulated via such structures.

As most of the SMRT-Cappable-seq TTSs are located 3′ of a transcribed gene, we hypothesized that these attenuator-like structures are also modulating read-through in the 3′ UTRs of transcripts after the polymerase has transcribed one or several gene(s). Under this hypothesis, attenuator-like structures are not only controlling the commitment of the RNA polymerase to transcribe operons but are also involved in a more fine-tune control of the gene compositions of operons.

To test this hypothesis, we studied the known *dapA-bamC* Rho-independent terminator that shows 40 and 11% differential read-through in M9 and Rich medium, respectively (Supplementary Fig. 7a), which is a differential read-through that we confirmed by qPCR. An attenuator structure was predicted 5′ of the TTS[28] (Supplementary Fig. 7b). We deleted the sequence predicted to be attenuator in the bacterial genome[29] and measured the level of read-through in both growth conditions using qPCR. We found that deleting the attenuator sequence, while it does not affect the read-through in M9 growth condition, it significantly (Unpaired *t*-test, $p$ value < 0.005) increased the read-through in Rich medium (Supplementary Fig. 7c and 7d). This indicates that this attenuator controls the conditional-regulated termination.

Taken together, these results demonstrate that a large number of operon variants are resulting from read-through at TTS. With the majority of operon termination sites having substantial read-through, this study redefines termination sites as a major point of control of gene expression. Here we show that the termination sites of operons have similar modular capacities to create polycistronic transcripts with variable gene compositions. Our results provide experimental evidences that a large fraction of

bacterial gene expression is regulated by conditional-regulated termination possibly via attenuator structures.

## Discussion

By highlighting a comprehensive view of the full-length transcriptional landscape, this study provides an important resource for functional annotation and the understanding of gene regulation in bacteria and microbiomes. Applications in microbiomes are particularly appealing since SMRT-Cappable-seq will be able to uncover partial or complete metabolic pathway by phasing functionally related genes on the same sequencing reads.

This unique ability to phase transcripts over long distances has also revealed the complexity of operon structures, complexity driven by both the regulation of transcription initiation and the modularity of transcription termination notably termination read-through. Studies have accumulated evidence of read-through for specific transcripts[21]. Consistent with our results, evidence of widespread read-through at 227 annotated *E. coli* termination sites has been shown in reporter constructs and the degree of such read-through can be affected by the extent of the polyU track[23]. Our study adds to the body of evidences for termination read-through by demonstrating the pervasive and modular nature of read-through across the native bacterial transcriptome. It is likely that in native context, the read-through is under the control of a number of additional factors such as the strength of the promoter that determines the degree of cooperation between RNA polymerase molecules in transcription elongation[30,31]. In turn, efficient translation promotes faster elongation and possibly higher read-through at termination sites. Consistent with this idea, we have experimental evidence that the degree of read-through at termination sites can be correlated with the TSS. It would be interesting to further study the mechanism leading to such correlation and whether or not the strength of the promoter is one determinant of read-through. This read-through represents a powerful mechanism to control elongation of transcription unit and include additional genes. In some cases, this control can be

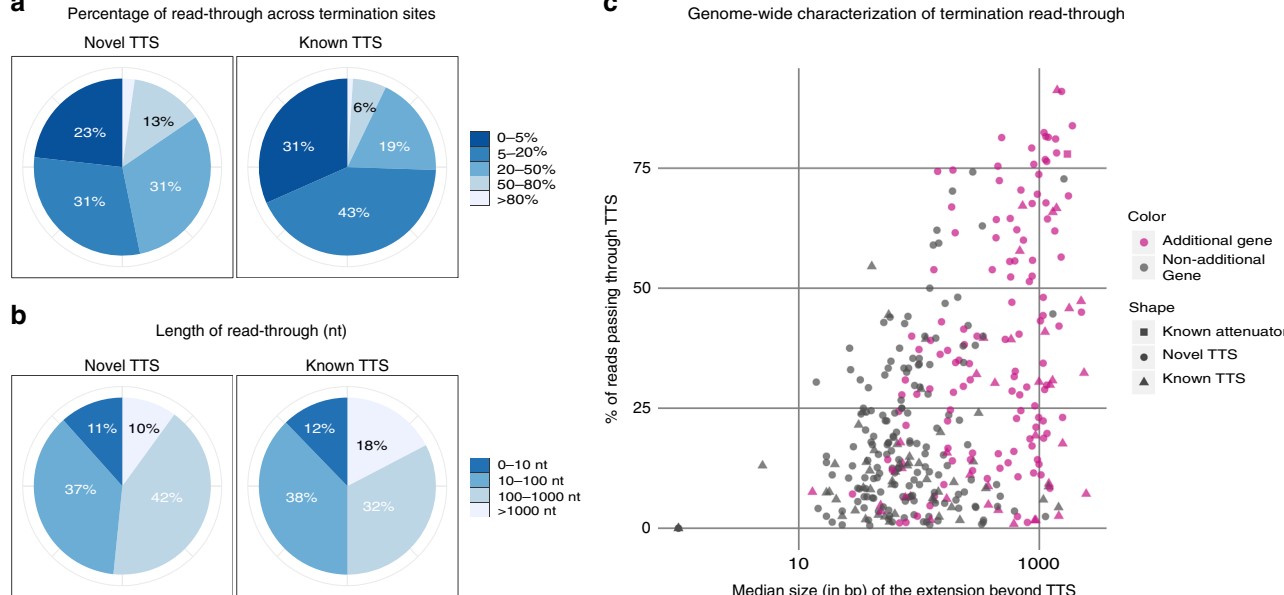

**Fig. 4** The vast majority of termination sites have read-through. **a** Pie chart showing the range of percentage of read-through for novel (left panel) and known (right panel) terminator sites in M9 medium. In 238 out of 310 novel TTS, the read-through is above 5% (Supplementary Data 1). For 67 of the 98 previously known TTS, the read-through is above 5%. **b** Pie chart showing the range of median size in bp of the 3′ extended region passing the termination site in M9 medium. 151 out of 408 termination sites contain additional genes. **c** Relation between the percentage of read-though (*y*-axis) and median size of the region (in bp) located 3′ of the TTS (*x*-axis) in M9 medium. TTSs with read-through containing additional gene(s) are in pink

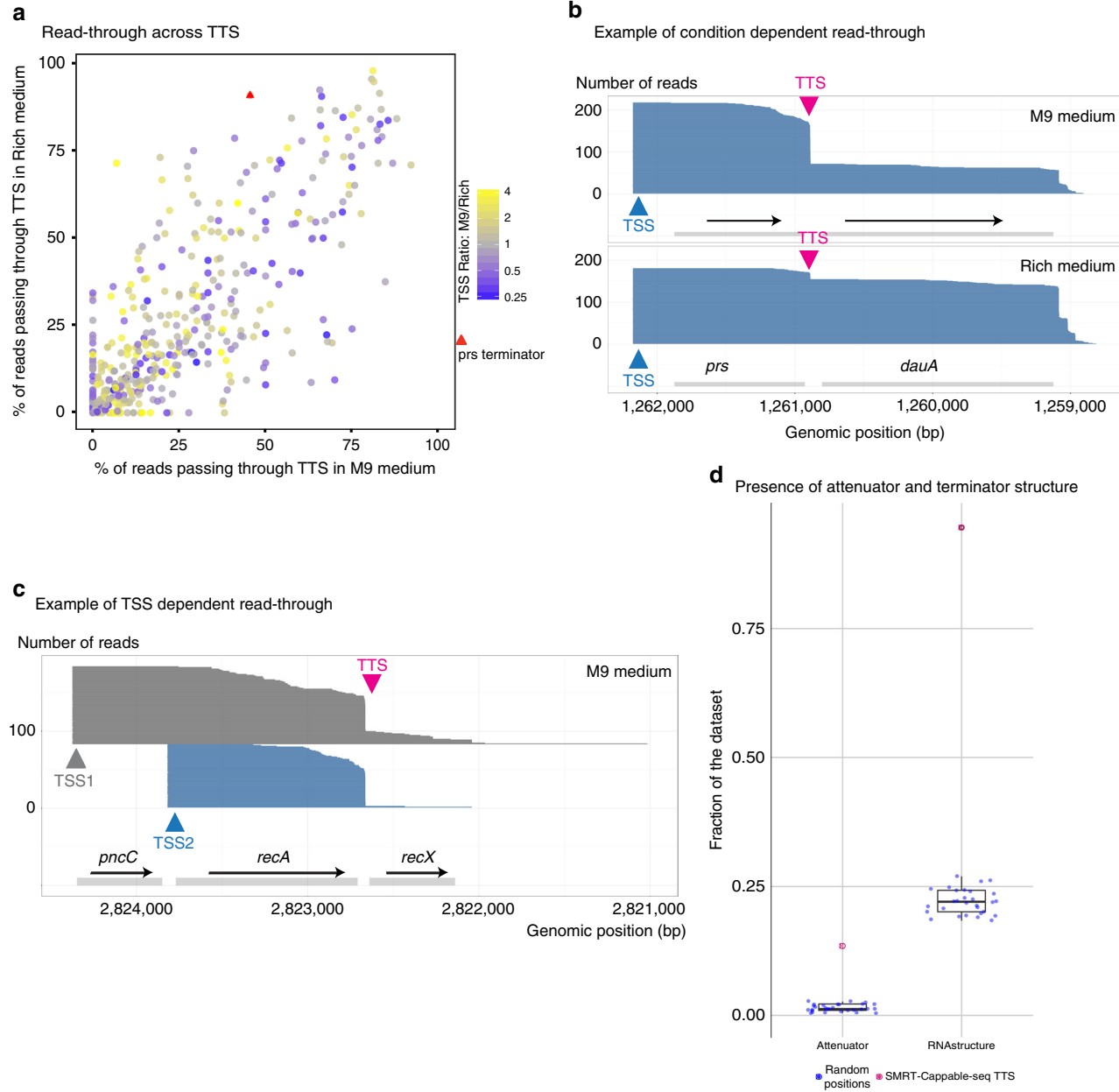

**Fig. 5** Condition dependent read-through and TSS-dependent read-through. **a** Percentage of reading-through across TTS in Rich and M9 conditions. Colors denote the ratio between expression levels of transcripts in M9 vs Rich medium at TSS. **b** An example of the *prs-dauA* operon containing a previously known TTS (red arrow), where the rate of read-through is condition dependent. The TTS between the *prs* and *dauA* gene shows 91% read-through in Rich medium as opposed to 46% read-through in M9 medium. As a result the *dauA* gene is up-regulated in Rich medium compared to M9 growth condition despite comparable expression level between conditions for its co-transcribed *prs* gene. **c** An example of a TSS-dependent read-through for overlapping transcripts in the same orientations. 32% read-through is observed for the long transcript (TSS1) vs 5% read-through for the short transcript (TSS2). **d** Percentage of termination sites in M9 medium (red) and random positions (blue) with predicted attenuator structures (left panel) (Pasific algorithm[28]) and predicted termination loop (right panel). 55 out of 408 termination sites have a predicted attenuator structure, and 386 of 408 termination sites have a predicted terminator structure

condition dependent and lead to the net up or down regulation of gene expression despite constitutive promoter activity.

Such modularity in the degree of transcription termination leads to a spectrum of sub-operons within larger operons and the ability of bacteria to transcribe genes in an array of TCs. In the case of RNA-seq these TCs are hidden and their importance has been underestimated. Yet, several studies have demonstrated that not only the level of gene expression but also the larger TC by which a gene is expressed is important for function. For example,

a recent study has shown that co-translation of genes on the same transcripts influences the proper arrangement and folding of complex protein structures[32], highlighting the importance of relating gene expression in the context of other genes on the same transcripts.

## Methods
**Growth conditions and RNA isolation for SMRT Sequencing**. *E. coli* k-12 strain MG1655 was grown at 37°C in M9 minimal medium with 0.2% glucose and Rich

medium (per liter 10 g Tryptone, 5 g Yeast Extract, 5 g sodium chloride, pH 7.2), respectively. The culture was grown to late log phase (OD600 between 0.55–0.6). Two volumes of RNAlater (Life Technologies) was added to the culture and saved at 4 °C overnight. The RNA was extracted using RNAeasy Midi kit (Qiagen). The isolated RNA had a RNA integrity number (RIN) above 9.0 as determined by Bioanalyzer (Agilent), and was used for SMRT-Cappable-seq.

**Enrichment of full-length prokaryotic primary transcripts**. The capping reaction was performed on total RNA: 5 μg of E. coli RNA was incubated in the present of 0.5 mM DTB-GTP (New England Biolabs) and 100 units of Vaccinia Capping Enzyme (New England Biolabs) and 0.25 units of yeast pyrophosphatase (New England Biolabs) for 0.5 h at 37 °C in 100 μl reaction volume. In order to measure the recovery of triphosphate transcripts, 1 ng of in vitro synthesized Gluc (Gaussia Luciferase) transcripts were mixed with the E. coli RNA for the capping and following reactions. The capped RNA was purified using AMPure beads.

Next, a polyA tail was added in vitro by incubating the capped RNA in 50 μl reaction volume with 20 units E. coli Poly(A) Polymerase (New England Biolabs) and 1 mM ATP for 15 min at 37 °C. The capped and tailed RNA was purified using AMPure beads and eluted in 50 μl low TE buffer. A volume of 10 μl of the reaction (non-enriched RNA) was put aside and used as control.

The capped RNA was enriched using hydrophilic streptavidin magnetic beads (New England Biolabs). A volume of 40 μl of beads were prepared by washing with Washing Buffer and suspended in 40 μl Binding Buffer (see below for composition). A volume of 40 μl of capped and tailed RNA was incubated with the beads at room temperature for 30 min on a rotator. The beads were then washed thoroughly three times with binding buffer and three times with washing buffer. To elute the RNA, the beads were resuspended in 26 μl Biotin Buffer and incubate at 37 °C for 25 min on a rotator, and 14 μl of Binding buffer was added and incubate for another 5 min. The Biotin eluted enriched RNA was purified with AMPure beads and eluted in 20 μl Low TE.

Low TE buffer: 1 mM Tris-HCl pH7.5; 0.1 mM EDTA
Binding Buffer: 10 mM Tris-HCl pH7.5; 2 M NaCl; 1 mM EDTA
Washing Buffer: 10 mM Tris-HCl pH7.5; 250 mM NaCl; 1 mM EDTA
Biotin Buffer: 1 mM Biotin; 10 mM Tris-HCl pH7.5; 0.5 M NaCl; 1 mM EDTA

**cDNA synthesis for SMRT Sequencing**. A volume of 20 μl of enriched RNA (SMRT-Cappable-seq) and 10 μl of non-enriched RNA (control)) were used in a 40 μl first strand cDNA synthesis reaction with 400 units of ProtoScript II Reverse Transcriptase (New England Biolabs) and 50 μM RT primer (Supplementary Table 1). Reactions were incubated at 42 °C for 1 h, then 50 units Rnase If (New England Biolabs) was added and incubated at 37 °C for another 30 min. The reactions were purified using AMPure beads and eluted in 20 μl Low TE.

The polyG was added to the 3′end of cDNA for second-strand synthesis using Terminal transferase (TdT, New England Biolabs). The purified cDNA/RNA duplex samples were incubated with 10 units TdT and 2 mM dGTP at 37 °C for 30 min. For the SMRT-Cappable-seq RNA, the reaction was purified using 30 μl hydrophilic streptavidin magnetic beads (New England Biolabs) as mentioned above but without Biotin buffer elution. The washed beads with the bound cDNA/RNA duplex were resuspended in 30 μl Low TE. For control RNA, the reaction products were purified using AMPure beads and eluted in 30 μl Low TE.

**SMRTbell library preparation and SMRT Sequencing**. We prepared two libraries using 5 μg of E. coli RNA from M9 and Rich medium, respectively. These two libraries were sequenced using PacBio RS II platform as described below (Supplementary Fig. 1).

Second-strand cDNA synthesis was performed on both control and enriched first strand cDNA/RNA duplex samples with LongAmp HotStart Taq DNA polymerase (New England Biolabs) using Pac_oligodC20 primer following the manufacturer's instructions.

PCR cycle number optimization was used to determine the optimal amplification cycle number for the downstream large-scale PCR reactions following the standard iso-seq protocol [http://www.pacb.com/wp-content/uploads/2015/09/User-Bulletin-Guidelines-for-Preparing-cDNA-Libraries-for-Isoform-Sequencing-Iso-Seq.pdf]. Second-strand cDNA was then bulk amplified with LongAmp HotStart Taq using Pac_for_dU and Pac_rev_dU primers, which contain an internal 5′-dUTP. Large-scale PCR products were purified with AMPure PB beads (Pacific Biosciences) and quality control was performed on a BioAnalyzer (Agilent). cDNA was then subjected to size fractionation using the Sage BluePippin system (Sage Science), collecting three size-bins: 1–2 kb, 2–3 kb, and 3–6 kb. Following size selection, size-selected cDNA was re-amplified using LongAmp HotStart Taq. Purified, size-selected cDNA was eluted in 20 μl Low TE and then digested using 1 unit USER (New England Biolabs) to create sticky ends. The sticky ended cDNA for each size-bin was then ligated to 1 μM hairpin adapters that contain complementary sticky ends to create SMRTbell templates using 2000 Units T4 DNA ligase (New England Biolabs). After incubation at room temperature for 1 h for ligation, 100 Units Exonuclease III (New England Biolabs) and 10 Units Exonuclease VII (USB) were added into the ligation reaction to remove the incompletely formed SMRTbell templates.

SMRTbell libraries for the 2–3 kb and 3–6 kb size-bins were subjected to an additional round of size selection on the Sage BluePippin to remove trace amounts of small inserts. Primer was annealed and samples were sequenced on the PacBio RS II (Menlo Park) for each respective size-bin. A total of 4 SMRTcells per size-bin were sequenced using P6-C4 chemistry and 4-h movies.

In addition, to measure the reproducibility, we prepared another two libraries using 5 μg of E. coli RNA from M9 medium as biological replicates, and sequenced them using the PacBio sequel platform.

For PacBio Sequel platform, we performed the same amplification and adapter ligation steps as mentioned for the RS II sequencing, except that the amplified cDNA was subjected to size selection using 0.4X AMPure beads to remove the small fragments below 1 kb. Primer was annealed and samples were sequenced on the PacBio Sequel (Menlo Park) following standard protocol. A total of 2 SMRTcells per replicate were sequenced using 8-h movies.

All the primers and adapter are listed in the Supplementary Table 1.

**qPCR**. SMRT-Cappable-seq enrichment: The first strand cDNAs were used as templates to determine the enrichment of primary transcript. The level of rRNA and mRNA was examined using qPCR (Luna Universal qPCR Master Mix, New England Biolabs) with primers targeting E. coli rrlH, rrlA, hupB, rmf genes, and the Gluc gene (Supplementary Table 1).

Deletion strains: To examine the degree of read-through across the dapA-bamC termination site, the cDNA was synthesized from RNA of wild type and deletion strain using ProtoScript II Reverse Transcriptase (New England Biolabs) with random primers following the manustructure's instructions. The degree of read-through was measured by comparing the expression level of RNA upstream of TTS (qPCR1) with the level downstream of TTS (qPCR2) (Supplementary Fig. 7b) using qPCR (Luna Universal qPCR Master Mix, New England Biolabs).

All the qPCR primer sequences are listed in the Supplementary Table 1.

**Genomic modifications**. To determine the effect of attenuator structure on transcription termination, we deleted a predicted attenuator region of the dapA-bamC terminator (Supplementary Fig. 7b) from E. coli genome (deletion strain DEL) and compared the read-through across this termination site. An E. coli B strain C2566 (New England Biolab), referred to in the text as "wild type," was used as the parental strain for genomic modification[29]. The wild type and deletion strain were grown in M9 and Rich grown condition late log phase and the RNA was extracted using RNAeasy Mini kit (Qiagen).

**Mapping to the NC_000913.3 genome**. The SMRT-seq CCS (circular-consensus) reads were processed using the pacbio_trim.py script. This script contains 2 functions: the filter and poly function. The filter function removes chimeric reads containing more than one AdaptorL (Supplementary Table 1) and reads without AdaptorL. The remaining reads were trimmed for 3′end polyA sequences and 5′ end polyC sequences using the poly function.

The trimmed reads were mapped to the E. coli genome using BLASR (RRID: SCR_000764) (-clipping soft). The alignment and all analyses were based on the E. coli genome NC_000913.3 and corresponding gene annotation expected for the rRNA analysis (Supplementary Note 3).

Additionally, we used adjust_3end.py script on the mapped reads to trim the remaining of polyA tails at the 3′end of the mapped reads.

The RNA-seq Illumina data downloaded from the European Nucleotide Archive (SRR3132588) were trimmed using Trimgalore [http://www.bioinformatics.babraham.ac.uk/projects/trim_galore/] and mapped to the E. coli genome NC_000913.3 using BWA (RRID:SCR_010910) version 0.7.5a-r418[33]. NCBI gene annotation from NC_000913.3 was used to define gene body and the gene expression level was estimated using bedtools (RRID:SCR_006646) multicov version v2.24.0 (-s parameter). For each gene, the number of Illumina reads was compared with the number of PacBio reads overlapping with the gene.

**TSS determination**. To identify TSS, we defined T as the number of reads in the same orientation starting at a given position and TPM (tag per million) as T divided by the number of mappable reads multiplying by one million. TPMs were calculated for each position on the genome. We calculated the SMRTratio, which is the ratio of SMRT-Cappable-seq TPM divided by Control TPM. For this analysis, a pseudo-count of 1 has been added to T at each position. Only positions with a SMRTratio above 1 were retained.

Nearby TSS positions in the same orientation within 5 bp distance were clustered together and only the position with the largest number of reads was used as the representative TSS position for the cluster. The number of reads at TSS corresponds to the sum of T in the cluster.

Only TSSs with more than four reads were retained in the final set of TSS. 2186 and 1902 TSS were found for the M9 and Rich medium growth condition, respectively. TSSs were considered as common between growth conditions if their genomic positions are identical and on the same strand. The pearson correlation of TSS between conditions is calculated by comparing TSS scores between M9 and Rich growth conditions.

The counting and clustering were performed using TSS_analysis.py count and cluster function.

The SMRT-Cappable-seq clustered TSSs under M9 growth condition were compared with the previous Cappable-seq results[2] as described in the Supplementary Note 5.

**TTS determination**. In this analysis, SMRT-Cappable-seq TTSs were defined as positions with significant accumulation of reads at 3′end. To determine the TTSs, first the number of transcripts with unique 3′ ends was counted for each clustered TSS using TSS_analysis.py count function. To distinguish true TTS from random read end due to incomplete transcription or degradation, we performed a statistical binomial test. More specifically, we calculated the the distance (D in bp) from the 3′ end of a read to its TSS. And if the percent (the number of reads ending at this position divided by the number of reads with the same TSS) was 200 times higher than the probability of random ending at this position (which is 1/D) and there were at least 10 reads, we defined this position as a TTS. The consecutive binomial probabilities were calculated using pbinom function in R, and one-tailed test at the 95% confidence level was used to delimit the statistically significant ends (TTS). The binomial test was performed using the binomial test.R script. In very few cases, TTS positions were within 5 bp. Therefore we merged these nearby TTSs into single site corresponding to the most 3′end position. The percentage of reads passing-through the TTS site was calculated as described in the Supplementary Note 7. TTS positions are provided in Supplementary Data 1.

The defined TTSs were compared with the previously experimentally identified termination sites and attenuator sites from Ecocyc (RRID:SCR_002433)[34] as described in the Supplementary Note 6.

**RNA structure analysis at transcriptional termination sites**. RNA sequence 40 bp upstream of the defined TTS was used for predicting secondary structure using RNAstructure[35] version 5.8.1. The RNA structure with the lowest free energy was predicted using the Fold function with the following parameters:–loop 30–maximum 20–percent 10–temperature 310.15–window 3. The RNA structure with free energy lower than −10 was defined as having a termination loop in Fig. 5d.

**Antiterminator prediction at transcriptional termination sites**. RNA sequence 100 bp upstream of the defined TTS was used to predict the antiterminator structure using PASIFIC webserver[28] with parameters optimized for detecting the attenuator. The RNA sequence showing a score above the 0.5 threshold was reported as having an attenuator in Fig. 5d.

**Motif logo analysis at TTS and TSS**. The −30 bp to 10 bp region of the defined TTS and the −45 bp to 5 bp region of the defined TSS under M9 condition was used for motif analysis. Motif logos were generated using the program weblogo3[36].

**Extended RegulonDB operons**. In this study, operons were determined as unique combination of genes included in individual transcripts. Therefore, reads fully covering the same genes were considered to be from the same operon.

The previously annotated operons in RegulonDB were compared with the SMRT-Cappable-seq reads from the combined M9 and Rich data set. If the read(s) fully covers an annotated operon and an additional gene(s), this annotated operon was extended by SMRT-Cappable-seq. Covering was determined using bedtools intersect version v2.24.0 (-s parameter).

**SMRT-Cappable-seq operons determination**. To determine the operon based on the SMRT-Cappable-seq result with high confidence, only transcripts with precise TSS (with over 4 reads) were used for this analysis as mentioned above. The start and end of the operon were defined by the TSS and TTS, respectively. For those transcripts without a defined TTS, the longest transcript was used for operon determination. Only the genes that were fully covered by SMRT-Cappable-seq transcripts were annotated into the operon.

Finally the SMRT-Cappable-seq defined operons from both M9 and Rich conditions were compared with the previous annotation in RegulonDB[16] (Supplementary Note 8). The defined SMRT-Cappable-seq operons are shown in Supplementary Data 2.

**Gene context analysis**. For each annotated gene, the number of operons that include the gene was calculated, and the gene contexts between M9, Rich condition, and RegulonDB (RRID:SCR_003499) were compared.

**Go enrichment analysis**. Go enrichment analysis and p value for genes with additional contexts in Rich medium was performed using the Gene Ontology Consortium[37]. The Bonferroni correction for multiple testing has been applied.

**Code availability**. The custom scripts used for the data analysis in this study are freely available on github [https://github.com/elitaone/SMRT-cappable-seq].

**Data availability**. Sequencing data and processed results were deposited and available at Gene Expression Omnibus GSE117273. All other data are available from the authors upon reasonable request.

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

## Acknowledgements
The authors would like to thank Rich Roberts, I. Schildkraut, L. Raleigh, M. Berkmen, B. Jack, T. Carlow, L. McReynolds, H. Runz, and D. Wang for critical comments.

## Author contributions
B.Y. developed the methodology and performed the experiments. M.B. and T.A.C. assisted with the library preparation and performed the SMRT sequencing. B.Y. and L.E. analysed the data. L.E. and B.Y. wrote the manuscript. All authors approved the final manuscript.

## Additional information

**Competing interests:** New England Biolabs Inc. and Pacific Biosciences of California, Inc supported this research. L.E. and B.Y. are employees of New England Biolabs. M.B and T.A.C. are employees of Pacific Biosciences of California.

