## [Peer Review File · Nature Communications]

Reviewers' comments:

Reviewer #1 (Remarks to the Author):

While I still feel that the datasets generated in this manuscript would be valuable to the scientific community, the authors have failed to adequately address any of my major concerns from the the previous review at [REDACTED]. For the manuscript to be acceptable for publication in Nature Communications, I think the authors will need to shorten the manuscript to report the key data (i.e. list of RNA ends, list of operons), compare their data in detail to those from reference 21, and incorporate at least one additional replicate dataset.

Here is a summary of the major comments from my previous review:

Comment #1: the concept of extensive terminator read-through has been described previously in reference 20.

- Reference 20 includes an analysis of 317 natural E. coli terminators, but the authors misrepresent this paper as looking only at synthetic terminators. Many of the 317 terminators show extensive read-through. While the authors of the current manuscript have somewhat tempered their conclusions about the novelty of their findings on read-through, this is still presented as a major discovery. I think the authors' data quantifying read-through in a native setting are an important addition to the literature, but the work of reference 20 should be more extensively discussed.

Comment #2: a previous study (reference 21) has shown that terminator read-through leads to complex operon arrangements.

- Reference 21 is the current "gold standard" in the field for operon structure in E. coli. The authors compare their data to RegulonDB. I don't think RegulonDB includes the data from reference 21. This would be a much more important comparison. I am confident that the data in the current study represent a more accurate and in-depth view of operon structure, but it is critical to compare the two studies before concluding this. Moreover, the novelty of this observation should be de-emphasized.

I compared the first 10 operons listed in Table S4 to those listed in reference 21. I was struck by two things. First, there is considerable overlap (4 of the 10 are listed in both studies). Second, one of the newly reported operons contains an entire gene in antisense. This phenomenon has been reported before in Listeria (PMID: 23268228). I think it would be very useful to report a list of such transcripts for E. coli.

Comment #3: the 3' ends identified in this study likely include many that are generated by RNA processing.

- The method used to define TTSs assumes that sites of RNA processing will be randomly distributed across RNAs, which is not the case. Hence, I think it is very likely that the TTS list includes many processed 3' ends. This is almost inevitable for RNAs that are terminated by Rho, since these 3' ends are rapidly processed. The authors compare their 3' ends to RNase III processing sites and find relatively little overlap. However, RNase III is one of many RNases in E. coli, with RNase E being the major RNase E for processing mRNAs. Hence, it is incorrect to conclude that "these results indicate that a minimal number of these TTSs are due to processing". Nonetheless, the authors' TTS list (which should probably be renamed since some of them won't be TTSs) is useful because it represents a list of upstream boundaries for the true TTSs.

Comment #4: the authors should expand their analysis of regulated read-through and TSS-dependent read-through.

- Regulated read-through and TSS-dependent read-through are very interesting phenomena that are highlighted in this study. While I agree that an in-depth analysis of these phenomena is beyond the scope of this study, it would be useful to show how frequently they are observed. For example, for regulated read-through, the authors could plot read-through rates in rich medium against those in minimal medium. Also, the authors should describe the deletion shown in Figure S6B. It is impossible to interpret these data without knowing the mutation made. They should also show the predicted RNA structure in the UTR.

Additional major comments:

- Without replicate datasets, I am hesitant to draw any solid conclusions from analyses such as the ones presented in Figure 3C and 4A. I had not previously appreciated how few sequence reads were acquired that map to non-rRNA regions. Because of this, I think the authors should repeat the experiment for at least one of the growth conditions, and show how reproducible replicate datasets are.

- Comparison of SMRT-Cappable-seq to an Illumina dataset needs a sample made from the same RNA. The currently presented data do not show a particularly good correlation between the methods, but this is likely due to the fact that the Illumina data come from an unrelated paper.

Additional comments:

- I still don't fully understand Figure 2B. I appreciate the point that the authors are making with this figure, but they should find a clearer way to represent this, or not show this figure at all. I think the text clearly conveys the message that many operons are extended using the SMRT-Cappable-seq data.

- The differences in transcription context numbers for different genes (Figure 3C) can easily be explained by differences in expression levels. More highly expressed genes are more likely to be detected in a higher number of contexts. This is especially true for the datasets presented here, which have relatively few reads (<500,000, most of which map to rRNA). This is almost certainly the case for genes involved in translation (see discussion at the top of page 6), which you would expect to be expressed more highly in rich medium than in minimal medium.

- The authors should remove Figure 5B. They refer to this as "transcription coordination", and while it is unclear what this term means, it is misleading. There are many examples of bidirectional terminators, but this is not such a terminator. I suspect it is purely coincidental that the two RNAs terminate at the same position.

- The authors have not adequately addressed my previous comment about the analysis shown in Figure 4D. Regulated attenuators (i.e. those that switch between terminated and anti-terminated states) are presumably strongly enriched for terminator stem loops structures, as are the 3' ends presented in the current study. Hence, it is expected that there would be an enrichment of regulated attenuator structures in the TTS list. The key question is whether TTSs whose read-through varies according to the growth conditions are even more enriched for regulated attenuator structures.

- The authors refer several times to "riboswitches", but they mean regulated attenuators.

Riboswitches are strictly defined by their ability to bind to small molecule ligands, resulting in a change in secondary structure. It's also worth noting that for many riboswitches, these structural changes impact translation rather than transcription termination. While many riboswitches are regulated attenuators, there are many other such RNA structures, including T-boxes, and the classical attenuators described for amino acid biosynthetic operons.

- 3' ends were mapped after addition of a polyG tail (or possibly a polyA tail – the figure and methods say "A" but the main text says "G"). Hence it is impossible to precisely map the 3' end for RNAs that naturally end in a G (or A, depending on the tail added). This is not a critical problem, but should be mentioned in the text. Additionally, the authors should describe how 3' ends were called in such cases.

- Figure 4D is cited out of order.

- I don't understand what Table S3 shows, especially in comparison to Table S4.

- It would be extremely helpful to provide a list of transcripts with start and stop coordinates, and the genes they cover. In other words, a combination of Tables S2 and S4.

- In Figure 1D, why are there reverse-strand reads in the ygaZ-emrB operon in the Illumina dataset?

Reviewer #2 (Remarks to the Author):

The authors provide a comprehensive survey of full-length, unprocessed transcripts of E.Coli MG1655 grown in rich and minimal media, using a protocol that allows effective capture of 5' and 3' termini of transcripts followed by PacBio SMRT sequencing. The data reveal that readthrough at many intrinsic terminators results in novel operon arrangements with many genes being expressed in as many as 15 transcriptional contexts. The authors also show that at least in one case a "riboswitch like structure" controls the efficiency of transcription termination depending on the growth conditions. Many genes with yet unknown function could now be assigned to extended operones, which may facilitate their functional analysis.

This is an important resource report performed at a high technical level. Most of the concerns raised with the previous version of the manuscript have been constructively addressed. The manuscript should be of interest to the broad readership of Nature Communication.

Minor issues:

Mapping PacBio and Illumina reads and estimating gene expression levels:

- no mention is made of read number normalization for the Illumina data (for both sequencing depth and gene length). Instead of comparing raw read numbers it would be more appropriate to compare abundance estimates.

- in the TSS determination algorithm, how was the cutoff value of 4 determined? It is clear how the scores were calculated but it would be useful to see the distribution of the calculated scores to reveal the "noise level" of this approach.

- the "low abundance of premature termination sites found in the 5'UTR" is surprising considering recent evidence demonstrating a high abundance of Rho-mediated terminators within 5'UTRs (PMID:27662085). Likewise, it has been shown that Rho-mediated termination is pervasively active on the antisense strand (PMID:28648779). The authors should at least mention these papers.

- I wonder whether there is any correlation between the TTS readthrough and promoter efficiency (promoter-proximal RNA yield). In principle, the stronger the promoter, the faster the elongation, due to the "cooperation" effect of multiple elongating RNAPs (PMID:12730602). The diminishing pausing (backtracking) at the termination site suppresses termination (PMID:10230402). Likewise, is there any correlation between the strength of SD and termination readthrough? Again, efficient translation promotes faster elongation due to its anti-backtracking effect (PMID:20413502). Of course, such potentially anti-termination conditions could be evolutionary compensated with stronger intrinsic terminators. Still, it would be interesting to investigate and maybe to discuss this issue.

- some typos in the text need to be corrected ("define" where "defined" should be used in multiple places)

Reviewer #3 (Remarks to the Author):

The manuscript by Yan et al. has been improved in response to the reviewers' comments and it does report interesting findings that deserve to see the light. However, there still remain issues with both presentation and science.

Grammatical errors are numerous, despite the assertions that the manuscript has been proofread. If the authors can't do this themselves, they should retain services of a professional editing company. Failing to do so is disrespectful to readers and reviewers. It would appear that different sections were written by different people, based on the frequency of errors. Capitalization is haphazard; e.g., legends of figures 2 and S3 have every word capitalized, others do not. Spelling is phonetic in some cases; e.g., sens in Fig. 5 captions. Many of these problems could be identified by running a spell check.

Science: as noted earlier, the authors' definition of a riboswitch is loose. A riboswitch is not a secondary structure that has an effect on gene expression, it is an RNA sequence that is able to undergo a conformational change upon binding of a ligand (hence a switch). Until a cognate ligand is identified, one cannot call an RNA a riboswitch; see any of the early Breaker papers. RNA structures could affect gene expression in different ways, and the authors should qualify their regulatory statements.

While this reviewer has no doubt that the authors are very well qualified in sequence data collection and analyses, their familiarity with the mechanisms of gene expression appears to be more limited. Softening conclusions is one way to address this issue.

Reviewer #1 (Remarks to the Author):

While I still feel that the datasets generated in this manuscript would be valuable to the scientific community, the authors have failed to adequately address any of my major concerns from the the previous review at [REDACTED]. For the manuscript to be acceptable for publication in Nature Communications, I think the authors will need to shorten the manuscript to report the key data (i.e. list of RNA ends, list of operons), compare their data in detail to those from reference 21, and incorporate at least one additional replicate dataset.

Here is a summary of the major comments from my previous review:

Comment #1: the concept of extensive terminator read-through has been described previously in reference 20.

- Reference 20 (PMID: 23727987) includes an analysis of 317 natural E. coli terminators, but the authors misrepresent this paper as looking only at synthetic terminators. Many of the 317 terminators show extensive read-through. While the authors of the current manuscript have somewhat tempered their conclusions about the novelty of their findings on read-through, this is still presented as a major discovery. I think the authors' data quantifying read-through in a native setting are an important addition to the literature, but the work of reference 20 should be more extensively discussed.

We have now corrected the misrepresentation and removed synthetic terminators from the text to avoid misunderstanding. As also pointed by the reviewers we have now made a clearer distinction between reporter constructs and native transcriptome to better define the context of our discovery and included this information in the discussion.

We have more extensively discussed the findings from reference 20 (PMID: 23727987) in the context of our finding and added the following paragraph to the discussion: "*Studies have accumulated evidence of read-through for specific transcripts {Stringer, 2014 #408} or using reporter constructs {Chen, 2013 #407}: consistent with our results, evidence of widespread read-through at 227 annotated E. coli termination sites has been shown in reporter constructs and the degree of such read-through can be affected by with the extent of the polyU track {Chen, 2013 #407}. Our study adds to the body of evidences for termination read-through by demonstrating the pervasive and modular nature of read-through across the native bacterial transcriptome.*"

Comment #2: a previous study (reference 21) has shown that terminator read-through leads to complex operon arrangements. - Reference 21 is the current "gold standard" in the field for operon structure in E. coli. The authors compare their data to RegulonDB. I don't think RegulonDB includes the data from reference 21. This would be a much more important comparison. I am confident that the data in the current study represent a more accurate and in-depth view of operon structure, but

it is critical to compare the two studies before concluding this. Moreover, the novelty of this observation should be de-emphasized. I compared the first 10 operons listed in Table S4 to those listed in reference 21. I was struck by two things. First, there is considerable overlap (4 of the 10 are listed in both studies).

We did the comparison between SMRT-Cappable-seq experiment and reference 21 (PMID:25006232). We have included the result in the text: “We also compared the SMRT-Cappable-seq operons with operons described in a previous study (PMID: 25006232) and found 1335 new operons from which 454 are extended in SMRT-Cappable-seq. ”. It is worth noting that the dataset described by the reviewer is based on short read sequencing, and thus, suffers from the same shortcomings of short reads sequencing: the inability to phase the TSS and TTS together. Thus the operon structure definition is based on algorithm prediction. We therefore found a large number of those predicted operons extended by SMRT-Cappable-seq.

Second, one of the newly reported operons contains an entire gene in antisense. This phenomenon has been reported before in *Listeria* (PMID: 23268228). I think it would be very useful to report a list of such transcripts for *E. coli*.

Sparked by the reviewer’s comments, we further examined operons containing an entire gene in antisense direction and added the result to the manuscript: “Finally, we also found many operons (165 in M9 and 146 in Rich) which contain entire genes in both the sense and the antisense direction (see Figure 5C for example of such operon). This result confirms the widespread presence of long antisense RNAs (asRNAs) in *E. coli* recently described for *Listeria* [PMID: 19448609].”

Comment #3: the 3’ ends identified in this study likely include many that are generated by RNA processing.

- The method used to define TTSs assumes that sites of RNA processing will be randomly distributed across RNAs, which is not the case. Hence, I think it is very likely that the TTS list includes many processed 3’ ends. This is almost inevitable for RNAs that are terminated by Rho, since these 3’ ends are rapidly processed. The authors compare their 3’ ends to RNase III processing sites and find relatively little overlap. However, RNase III is one of many RNases in *E. coli*, with RNase E being the major RNase E for processing mRNAs. Hence, it is incorrect to conclude that “these results indicate that a minimal number of these TTSs are due to processing”. Nonetheless, the authors’ TTS list (which should probably be renamed since some of them won’t be TTSs) is useful because it represents a list of upstream boundaries for the true TTSs.

In *E. coli*, primary transcripts that retain a 5’ triphosphate are resistant to degradation because the major endonuclease RNase E is sensitive to 5’ structure, strongly favoring degradation of processed RNAs with a 5’ -monophosphate (Celesnik et al., 2007: Initiation of RNA decay in *Escherichia coli* by 5’ pyrophosphate removal). Degradation is stimulated by conversion of the

protective triphosphate 5' -end structure to a monophosphate by the pyrophosphate hydrolyase RppH (Deana et al., 2008). As Cappable-seq specifically capture triphosphate RNA, we expected to markedly enriched for primary transcripts that have also retained their original TTS.

Comment #4: the authors should expand their analysis of regulated read-through and TSS-dependent read-through.

- Regulated read-through and TSS-dependent read-through are very interesting phenomena that are highlighted in this study. While I agree that an in-depth analysis of these phenomena is beyond the scope of this study, it would be useful to show how frequently they are observed. For example, for regulated read-through, the authors could plot read-through rates in rich medium against those in minimal medium. Also, the authors should describe the deletion shown in Figure S6B. It is impossible to interpret these data without knowing the mutation made. They should also show the predicted RNA structure in the UTR.

TSS-dependent read-through: We have confidently identify TSS-dependent read-through for only a handful of cases. We have clarified the extend of TSS-dependent read-through by adding the following sentence : “.. we found for a handful of cases that the degree of read-through of a given terminator can vary according to the TSS, a phenomena that we term “TSS-dependent read-through” because the level of TTS read-through is correlated with the location of the transcript TSS.”

Regulated read-through: we have now added Figure 5A, plotting the correlation between TTS read-through in M9 versus Rich growth conditions as suggested by the reviewer.

Concerning the deletion: we have now added the information about the deletion in Figure S6B.

Additional major comments:

- Without replicate datasets, I am hesitant to draw any solid conclusions from analyses such as the ones presented in Figure 3C and 4A. I had not previously appreciated how few sequence reads were acquired that map to non-rRNA regions. Because of this, I think the authors should repeat the experiment for at least one of the growth conditions, and show how reproducible replicate datasets are.

As suggested by the reviewer, we have added the replicate analysis experiment (See text and Supplementary Figure S1): We examined the reproducibility of two technical replicates (under M9 condition) by comparing the gene expression level and the number of reads at TSS and TTS. In all metrics, these two technical replicates showed a Pearson's r correlation of >0.95 (p.value < 2.2e-16) for gene expression, TSS and TTS usage (Figure S1A, B, C), which indicates that our method is very

reproducible. We have added the following text to the manuscript: “In order to evaluate the reproducibility of the method, we have also performed a technical replicate (Material and Methods) and obtained a very good correlation between replicates in terms of gene expression (**Figure S1A**), TSS and TTS identification and quantification (**Figure S1B and C**).” and added Figure S1A, B and C to the manuscript.

Figure S1A, B, C

- Comparison of SMRT-Cappable-seq to an Illumina dataset needs a sample made from the same RNA. The currently presented data do not show a particularly good correlation between the methods, but this is likely due to the fact that the Illumina data come from an unrelated paper.

We actually have a fairly good correlation between Illumina RNA-seq and SMRT-Cappable-seq (Spearman's rank correlation 0.798) given that [1] the SMRT-Cappable-seq library has been size selected [2] SMRT-Cappable-seq selects for primary transcripts (containing a 5' triphosphate) in SMRT-Cappable-seq library and [3] RT-bias. Nanopore has recently published its native RNA sequencing technology and show similar correlations with Illumina RNA-seq (<https://www.nature.com/articles/nmeth.4577>).

Additional comments:

- I still don't fully understand Figure 2B. I appreciate the point that the authors are making with this figure, but they should find a clearer way to represent this, or not show this figure at all. I think the text clearly conveys the message that many operons are extended using the SMRT-Cappable-seq data.

We are now modified the figure legend to better explain figure 2B : “Each line corresponds to the size of the operons (in bp) with the previously annotated operons in pink and the extended operons in blue. Positions are relative to the 5' end of the annotated operon with 0 being the TSS of the annotated operons. In total there are 883 RegulonDB operons extended by SMRT-Cappable-seq from either 5' or 3' end or both.”

- The differences in transcription context numbers for different genes (Figure 3C) can easily be explained by differences in expression levels. More highly expressed genes are more likely to be detected in a higher number of contexts. This is especially true for the datasets presented here, which have relatively few reads (<500,000, most of which map to rRNA). This is almost certainly the case for genes involved in translation (see discussion at the top of page 6), which you would expect to be expressed more highly in rich medium than in minimal medium.

We agree with the concern of the reviewer that the difference in the number of transcriptional context between Rich and M9 could be the result of differences in expression levels. To address this concern, we plotted the $\log_2(\text{ratio})$ in expression between Rich and M9 condition compared to the difference in the number of contexts between Rich and M9 condition. We have attached the result of the analysis to this reply (figure below) and essentially found a minor correlation ($\text{corr} = 0.17$) between the two metrics. We therefore concluded that the difference in expression has only a small correlation with the difference in transcriptional context.

Correlation between the ratio of expression in Rich and M9 medium and the difference in number of context in Rich versus M9 .

- The authors should remove Figure 5B. They refer to this as “transcription coordination”, and while it is unclear what this term means, it is misleading. There are many examples of bidirectional terminators, but this is not such a terminator. I suspect it is purely coincidental that the two RNAs terminate at the same position.

We did not remove this figure because it is an interesting locus for several reasons :

[1] The termination site is shared between the sense and antisense : The sequence between the two termination site is 36 bp long and correspond to the following sequence : 5'GTGTGAAGTTGCCGGATGTGTTGCATCCGGCATGGC3'. We run the RNA loop structure prediction algorithm (<https://rna.urmc.rochester.edu/RNAstructureWeb/>) and found the same TTS loop structure in both sense and antisense orientation. This result suggests that the same terminator is used in both direction.

[2] It presents a new case of operon containing an entire gene in antisense direction (as rightfully pointed out by the reviewer as interesting excludon in his/her comment above).

To address the reviewer's comment regarding the term "transcription coordination", we changed the term to "TSS-dependent read-through" a term used by reviewer 1 that better describes this phenomena. We have therefore adjusted the manuscript to reflect the changes : "... we found, for a handful of cases that the degree of read-through of a given terminator can vary according to the TSS, a phenomena that we term "TSS-dependent read-through" because the level of TTS read-through is correlated with the location of the transcript TSS.

- The authors have not adequately addressed my previous comment about the analysis shown in Figure 4D. Regulated attenuators (i.e. those that switch between terminated and anti-terminated states) are presumably strongly enriched for terminator stem loops structures, as are the 3' ends presented in the current study. Hence, it is expected that there would be an enrichment of regulated attenuator structures in the TTS list. The key question is whether TTSs whose read-through varies according to the growth conditions are even more enriched for regulated attenuator structures.

Figure 4D is moved to be Figure 5E in the modified manuscript as suggested by the reviewer. We have not evaluated enough conditions to be able to clearly identify structural differences between condition-dependent attenuators versus condition-independent attenuators. Rich and minimal growth conditions are only two out of a large number of conditions for which *E. coli* can modify readthrough through terminator stem loops structures. Examining only these two conditions, we did not found significant differences likely for the reason stated above.

- The authors refer several times to "riboswitches", but they mean regulated attenuators. Riboswitches are strictly defined by their ability to bind to small molecule ligands, resulting in a change in secondary structure. It's also worth noting that for many riboswitches, these structural changes impact translation rather than transcription termination. While many riboswitches are regulated attenuators, there are many other such RNA structures, including T-boxes, and the classical attenuators described for amino acid biosynthetic operons.

We have now changed the riboswitches to regulated attenuators throughout the manuscript.

- 3' ends were mapped after addition of a polyG tail (or possibly a polyA tail – the figure and methods say “A” but the main text says “G”). Hence it is impossible to precisely map the 3' end for RNAs that naturally end in a G (or A, depending on the tail added). This is not a critical problem, but should be mentioned in the text. Additionally, the authors should describe how 3' ends were called in such cases.

We added 2 polynucleotides tails: A polyA tail is added to the 3'end of RNA for RT reaction. After RT, a polyG tail is added to the 3'end of the cDNA for second strand synthesis (equals to adding a polyC to the RNA sequence). We have this information in the method and Supplementary Figure 7. In addition, we have considered this problem during our data analysis: We have trimmed the polyG and polyA tail before the mapping step. For example if the original read containing the Adapter and poly tail sequence is:

```
TTGTACACTCTGTCGCTACGTAGATAGCGTTGAGTGCCCCCCCCCCCCCCCCCTGGAT.....GAAATCATAGC  
CCTGATTAATAAAAAAAAAAAAAAAAAAAAAAAAAAAAAAAAAAAAAAAAAAAGATCGGAAGAGCAC  
ACGTCTGAACTCCAGTCACAC
```

it will be trimmed to the following sequence before mapping:

```
TGGAT.....GAAATCATAGCCCTGATT
```

Therefore the transcript will end at the first non-A position at 3'end if it naturally ends in A.

As mentioned in the manuscript, we provide the details about the data analysis including this read trimming in the README file which are available at github (<https://github.com/elitaone/SMRT-cappable-seq>).

- Figure 4D is cited out of order.

We have fixed the out of order and moved Figure 4D to Figure 5E

- I don't understand what Table S3 shows, especially in comparison to Table S4.

Table S3 corresponds to the SMRT-Cappable-seq operons. Table S4 corresponds to RegulonDB operons that have been extended by SMRT-Cappable-seq. We have added Table S4 in response to a previous reviewer's comment. We have now changed the title of the Tables to clarified this misunderstanding : Supplementary table 3: SMRT-Cappable-seq operons. Supplementary table 4: RegulonDB operons extended by SMRT-Cappable-seq.

- It would be extremely helpful to provide a list of transcripts with start and stop coordinates, and the genes they cover. In other words, a combination of Tables S2 and S4.

We agree with the reviewer that the coordinates (with define start and end) should be complemented with genes covered by the transcripts. We have therefore added in Table S2 a column (column 13) containing all the genes that are fully covered by the transcripts.

- In Figure 1D, why are there reverse-strand reads in the ygaZ-emrB operon in the Illumina dataset? The RNA-seq Illumina data downloaded from the European Nucleotide Archive (SRR3132588). This dataset has been produce using a non-directional RNA library construction procedure. Hence, the more or less equal distribution of forward and reverse strand reads.

Reviewer #2 (Remarks to the Author):

The authors provide a comprehensive survey of full-length, unprocessed transcripts of E.Coli MG1655 grown in rich and minimal media, using a protocol that allows effective capture of 5' and 3' termini of transcripts followed by PacBio SMRT sequencing. The data reveal that readthrough at many intrinsic terminators results in novel operon arrangements with many genes being expressed in as many as 15 transcriptional contexts. The authors also show that at least in one case a "riboswitch like structure" controls the efficiency of transcription termination depending on the growth conditions. Many genes with yet unknown function could now be assigned to extended operones, which may facilitate their functional analysis.

This is an important resource report performed at a high technical level. Most of the concerns raised with the previous version of the manuscript have been constructively addressed. The manuscript should be of interest to the broad readership of Nature Communication.

Minor issues:

Mapping PacBio and Illumina reads and estimating gene expression levels:

- no mention is made of read number normalization for the Illumina data (for both sequencing depth and gene length). Instead of comparing raw read numbers it would be more appropriate to compare abundance estimates.

We have now updated the Figure 1C to reflect the Reads Per Kilobase of transcript, per Million mapped reads (RPKM) values instead of the raw read counts. The Spearman's rank correlation is 0.798 ($p.value < 2.2e-16$).

- in the TSS determination algorithm, how was the cutoff value of 4 determined? It is clear how the scores were calculated but it would be useful to see the distribution of the calculated scores to reveal the "noise level" of this approach.

We have decided on the cutoff 4 based on the fraction of 5' end reads that are mapping to a known TSS. Below is the curve showing the fraction of 5' end reads that are mapping to known TSS function of the cutoff. At cutoff 4 (red point), 93% of the 5' end reads are within 5 bp of a known TSS. We can add this information to the manuscript if the reviewer feels it would be beneficial.

- the “low abundance of premature termination sites found in the 5'UTR” is surprising considering recent evidence demonstrating a high abundance of Rho-mediated terminators within 5'UTRs (PMID:27662085). Likewise, it has been shown that Rho-mediated termination is pervasively active on the antisense strand (PMID:28648779). The authors should at least mention these papers.

To fully take advantage of the long read sequencing technology, SMRT-Cappable-seq libraries were size selected for large transcripts above 1kb. Thus, it is very likely that most of the small transcripts resulting from premature terminations at the 5' end of genes were eliminated during this size selection step. We have now added an explanation sentence to the manuscript to clarify this difference and added the first reference suggested by the reviewer: *“The relatively low abundance of premature termination sites in SMRT-Cappable-seq libraries is contrasting with the large number of known regulatory sites controlling transcript elongation in the 5'UTR of genes (PMID:27662085) {Santangelo, 2011 #370}. This inconsistency can be explained by the size selection step during library preparation that depletes for short transcripts characteristic of premature termination. We nonetheless identify premature transcripts derived from well known examples of termination sites in the 5'UTR region of the operon such as the Leu operon (Figure S3).”*

- I wonder whether there is any correlation between the TTS readthrough and promoter efficiency (promoter-proximal RNA yield). In principle, the stronger the promoter, the faster the elongation, due to the “cooperation” effect of multiple elongating RNAPs (PMID:12730602). The diminishing pausing (backtracking) at the termination site suppresses termination (PMID:10230402). Likewise, is there any correlation between the strength of SD and termination readthrough? Again, efficient translation promotes faster elongation due to its anti-backtracking effect (PMID:20413502). Of course, such potentially anti-termination conditions could be evolutionary compensated with stronger intrinsic terminators. Still, it would be interesting to investigate and maybe to discuss this issue.

We have added to the discussion this interesting point raised by the reviewer: “It is likely that in native context, the readthrough is under the control of a number additional factors such as, for example the strength of the promoter that determine the degree of cooperation between RNA polymerase molecules in transcription elongation (PMID:12730602). In turn, efficient translation promotes faster elongation and possibly higher readthrough at termination sites. Consistent with this idea, we have experimental evidence that the degree of readthrough at termination sites can be correlated with the TSS : it would be interesting to further study the mechanism leading to such correlation and whether or not the strength of the promoter is one determinant of readthrough.”

- some typos in the text need to be corrected ("define" where "defined" should be used in multiple places)

We have carefully proofread the manuscript and ensure consistency in spelling, abbreviations and capitalization.

Reviewer #3 (Remarks to the Author):

The manuscript by Yan et al. has been improved in response to the reviewers’ comments and it does report interesting findings that deserve to see the light. However, there still remain issues with both presentation and science.

Grammatical errors are numerous, despite the assertions that the manuscript has been proofread. If the authors can’t do this themselves, they should retain services of a professional editing company. Failing to do so is disrespectful to readers and reviewers. It would appear that different sections were written by different people, based on the frequency of errors. Capitalization is haphazard; e.g., legends of figures 2 and S3 have every word capitalized, others do not. Spelling is phonetic in some cases; e.g., sens in Fig. 5 captions. Many of these problems could be identified by running a spell check.

In response to this comment, we have now ensure consistency in the spelling and capitalization of words and run a spell-check over the document.

Science: as noted earlier, the authors' definition of a riboswitch is loose. A riboswitch is not a secondary structure that has an effect on gene expression, it is an RNA sequence that is able to undergo a conformational change upon binding of a ligand (hence a switch). Until a cognate ligand is identified, one cannot call an RNA a riboswitch; see any of the early Breaker papers. RNA structures could affect gene expression in different ways, and the authors should qualify their regulatory statements.

We agree with the reviewer. In our analysis, we used a published algorithm for the prediction of riboswitches that are consisting of a typical regulated attenuator. Accordingly, we have now changed the riboswitches to regulated attenuators throughout the manuscripts.

While this reviewer has no doubt that the authors are very well qualified in sequence data collection and analyses, their familiarity with the mechanisms of gene expression appears to be more limited. Softening conclusions is one way to address this issue.

As suggested by the reviewer, we have now rewritten sections to soften the conclusion, and we have also more extensively discussed the previous findings in the discussion.

REVIEWERS' COMMENTS:

Reviewer #1 (Remarks to the Author):

The authors have made several improvements to the manuscript. In particular, a replicate dataset is an important addition, as it demonstrates that the data quality is high. Many of the issues with the text and figures have been resolved. The authors have also included discussion of two particularly relevant papers. Nonetheless, I think there are still a lot of changes that need to be made to the text. Most of the changes the authors have made to the text are additions, leaving much of the problematic text in place.

Major comments:

1. The authors have added a comparison of their data to those from Conway et al., the most relevant published dataset. This is an important addition. However, the manuscript focuses almost exclusively on the comparison to RegulonDB (the term "RegulonDB" is mentioned 26 times in the manuscript!). In doing so, the authors overstate the significance of their work. In particular, the statement "This percentage represents a remarkable large number of new operons considering decades of studies done in *E. coli*" (bottom of pg. 4) is misleading, since it does not include a comparison to the most relevant published dataset. The authors also use an example of an extended operon (*tff-rpsB-tsf*) that is not novel, having been previously described by Conway et al., and they describe an operon linking *yedJ* and *yedR* to *dcm* and *vsr*, despite the fact that Conway et al. described an operon containing *yedJ*, *dcm* and *vsr*. Comparison to the Conway et al. paper should be presented first and throughout; comparison to RegulonDB should be mentioned briefly afterwards, and perhaps not at all. Specific examples that are not novel should be removed altogether. I suggest mentioning up-front why the new operon list is better than the list inferred from short-read data, since this highlights the importance of the new approach.
2. While I agree with the authors that capturing triphosphorylated RNA will help avoid sequencing of RNase E-cleaved transcripts, not all RNase E cleavage is RppH-dependent, and there are exonucleases that degrade from the 3' end, especially for Rho-terminated RNAs. See the recent paper from Dar and Sorek (NAR, 2018) that shows the extent of cleavage of Rho-terminated transcripts by PNPase and RNase II. There is no doubt that the "TTS" list includes processed ends. This is not a critical flaw, but should be acknowledged. The authors should soften the statement "capturing of the 5' triphosphate should also ensure the removal of degraded and/or processed transcripts on the 3' end" (pg. 2), limiting this de-enrichment to some RNase E processing, and mentioning that additional nucleases play important roles in RNA processing in *E. coli*, particularly for Rho-terminated transcripts. The paragraph later on (pg. 3-4) that discusses RNA processing should also be modified.
3. The authors use two programs to predict intrinsic terminators (Fig. 5E). I'm not familiar with these programs, but if they only rely on predicted structure, and not on structure followed by a U-rich sequence, then the discussion of these data should be softened. Finding a structured 3' end is expected of both true TTSs (at least for Rho-independent termination) and processed 3' ends (see Dar and Sorek, 2018 for more details on this for Rho-terminated RNAs), and hence should not be used as an indication that most TTSs are genuine. Lastly, a structured 3' end without a U-tract should not be referred to as a "termination structure" (pg. 4). If the programs look for a structure followed by a U-rich sequence, "termination structure" is appropriate.
4. The authors have not done enough to convince me that increases in transcriptional context (TC) for translation-associated genes in rich media are not due to expression increases (pg. 6, para. 3). The graph presented in the point-by-point response clearly shows that genes with higher TC in rich medium are more highly expressed than in minimal medium. For example, I count 49 genes with a TC value in rich medium that is at least 2 higher than the TC value in minimal medium, and only 3 of these 49 genes are expressed less in rich medium than in minimal medium. As an aside, I was confused by this graph since it appears to show genes with TC differences that are non-integer values. I recommend removing the third paragraph of pg. 6 that describes this phenomenon.
5. The authors have kept Fig. 5C, which shows transcripts on opposite strands terminating at the

same site. I strongly disagree that this is an interesting finding. While I am a little confused as to whether the termination site itself is shared, as opposed to the terminator hairpin (I assume it's the former, but the point-by-point response refers to a sequence "between the sense and antisense"), neither of these would be unexpected. Bidirectional terminators are widespread, and lead to termination at different sites, but use the same hairpin. Terminating at the same nucleotide position but on opposite strands is not the same as TSS-dependent differences in read-through. The TSSs may be different, but so is the rest of the transcript! There is no reason to expect the level of read-through to be the same for the sense and antisense transcripts. It's fine to use this as an example of an excludon, but it is not relevant for TSS-dependent read-through effects.

6. Pg. 8, first full paragraph. As I mentioned in my previous reviews, you would expect to find an enrichment for predicted attenuators in the TTS list because the TTSs are enriched for hairpins. The authors cannot tell whether TTS read-through is connected to predicted attenuator structure, so this paragraph should be removed.

Additional comments:

1. For Fig. S6, I still think it would be helpful to include the predicted structure(s). Note that there would be two structures if the authors' attenuation model is correct.
2. The authors still refer to riboswitches inappropriately. At the top of pg. 8 they describe a program they used to find attenuators, not riboswitches as they suggest. For the paragraph at the bottom of pg. 7/top of pg. 8, there is no need to focus on riboswitches. It would be more appropriate to describe attenuators. Not all riboswitches regulate transcription, and not all attenuators are associated with riboswitches.
3. Pg. 2, second full paragraph. The authors should mention here that transcripts ending in A will have their TTSs miscalled. This does not impact any of the conclusions, but is important for readers to know, especially if they plan to use the TTS list for downstream analyses.
4. Top of pg. 5, "found 1335 (56%) new operons from which 454 are extended by SMRT-Cappable-seq". I was confused by this – what is the difference between finding a new operon and extending a newly found operon?

Reviewer #2 (Remarks to the Author):

The authors have constructively addressed all my concerns and I believe this paper is suitable for publication in its present form. The only minor improvement I suggest is to replace ref 26 with PMID: 18158897. Ref 26 used single molecule approach to argue that 3 different intrinsic terminators they studied worked by different mechanisms (forward translocation, RNA shearing, and allosteric mechanism). This is highly unlikely and bulk of biochemical evidence argues that the only mechanism of intrinsic termination is the one described in PMID: 18158897. Conceptually similar "allosteric" mechanism also works for Rho-mediated termination (PMID: 20075920).

Reviewer #3 (Remarks to the Author):

This manuscript presents data that complement and extend the existing information. The ability to map the end of the transcript that starts kilobases upstream is unique and cannot be overstated. It is likely that the majority of the determined transcript ends correspond to primary termination

sites. The complexity of operon structures that was largely overlooked because of the short sequencing reads is quite striking.

The manuscript has been significantly improved, the results will be of interest to many research groups, and are easy to follow after the revisions. I have a few minor points for consideration: 5' UTRs: it is actually shocking that, given the size selection, the authors successfully identified any early TTS. Perhaps it would be helpful to state more directly that 5' UTR are typically very short, below the selected 1 kb size.

Factors affecting TSS-dependent readthrough: I thought this came up before, but this could have been a while. In addition to the strength of the promoter, which I am not sure will contribute that strongly because few promoters are strong enough (cooperation was shown for *rrn* promoters), the initial transcribed sequence (obviously different in each case) has been shown to have a direct effect on termination (PMID: 2467004). One could look whether transcribed regions that increase readthrough have a potential to base pair with the terminator stem; basically, very distant antiterminators.

5' nucleotide: it may be more appropriate to cite recent Belasco papers (e.g. Luciano 2017) instead of Deana 2008 as they have information. Although this is not critical, mentioning alternative 5' caps could ass interest.

Termination: Santangelo review is outdated; perhaps cite PMID 27023849 instead, or primary references.

REVIEWERS' COMMENTS:

Reviewer #1 (Remarks to the Author):

The authors have made several improvements to the manuscript. In particular, a replicate dataset is an important addition, as it demonstrates that the data quality is high. Many of the issues with the text and figures have been resolved. The authors have also included discussion of two particularly relevant papers. Nonetheless, I think there are still a lot of changes that need to be made to the text. Most of the changes the authors have made to the text are additions, leaving much of the problematic text in place.

Major comments:

1. The authors have added a comparison of their data to those from Conway et al., the most relevant published dataset. This is an important addition. However, the manuscript focuses almost exclusively on the comparison to RegulonDB (the term "RegulonDB" is mentioned 26 times in the manuscript!). In doing so, the authors overstate the significance of their work. In particular, the statement "This percentage represents a remarkable large number of new operons considering decades of studies done in *E. coli*" (bottom of pg. 4) is misleading, since it does not include a comparison to the most relevant published dataset. The authors also use an example of an extended operon (tff-rpsB-tsf) that is not novel, having been previously described by Conway et al., and they describe an operon linking yedJ and yedR to dcm and vsr, despite the fact that Conway et al. described an operon containing yedJ, dcm and vsr. Comparison to the Conway et al. paper should be presented first and throughout; comparison to RegulonDB should be mentioned briefly afterwards, and perhaps not at all. Specific examples that are not novel should be removed altogether. I suggest mentioning up-front why the new operon list is better than the list inferred from short-read data, since this highlights the importance of the new approach.

We have now removed the tff-rpsB-tsf operon as an example of a new extended operon and added another example of a new extended operon (one that was not found by Conway *et al.* Fig. 2b). We have conserved the tff-rpsB-tsf operon as an example of operon containing genes of various functions and added the fact that this operon has been described by Conway et al. in the manuscript: "*Furthermore the tff-rpsB-tsf-pyrH-frr transcript has been predicted in a previous study (PMID: 25006232) consistent with the existence of a functional and conserved extended operon containing all 4 protein coding genes.*" As suggested, we have removed the following statement: "*This percentage represents a remarkable large number of new operons considering decades of studies done in E. coli.*"

We also compared the SMRT-Cappable-seq operons with operons predicted in Conway *et al.*, and added the comparison in the manuscript. "*We also compared the SMRT-Cappable-seq operons with operons predicted in a previous study (PMID: 25006232), and we found around 40% of our operons were not predicted (see Fig. 2b for example).*" Many of these operons are due to the different internal TSSs (Supplementary Fig. 5 for examples).

Importantly, we have removed the term novel whenever inappropriately used and instead specify the dataset used. So for example instead of stating "We found 2347 operons are encoding unique combination of genes and amongst those, 840 (36 %) are novel" we state " *Thereby we defined 2347 operons that encode unique combination of genes, amongst which 840 (36%) are not annotated in the RegulonDB database*".

We did not however remove the comparison with RegulonDB: RegulonDB is a repository of decades of careful studies of operons and transcription control in E.coli. Most of the evidences for TSS, TTS and operons are derived from carefully designed experiments at single gene level, and it has been used by others (including Conway *et al.*) as the primary reference database for operons in bacteria.

And as suggested, we have highlighted the importance of our approach, which is the phasing of TSS and TTS, in the manuscript.

2. While I agree with the authors that capturing triphosphorylated RNA will help avoid sequencing of RNase E-cleaved transcripts, not all RNase E cleavage is RppH-dependent, and there are exonucleases that degrade from the 3' end, especially for Rho-terminated RNAs. See the recent paper from Dar and Sorek (NAR, 2018) that shows the extent of cleavage of Rho-terminated transcripts by PNPase and RNase II. There is no doubt that the "TTS" list includes processed ends. This is not a critical flaw, but should be acknowledged. The authors should soften the statement "capturing of the 5' triphosphate should also ensure the removal of degraded and/or processed transcripts on the 3' end" (pg. 2), limiting this de-enrichment to some RNase E processing, and mentioning that additional nucleases play important roles in RNA processing in E. coli, particularly for Rho-terminated transcripts. The paragraph later on (pg. 3-4) that discusses RNA processing should also be modified.

We have modified the sentence to acknowledge the possible existence of processed 3' end in our dataset (see underlines): "*The capturing of the 5' triphosphate is expected to markedly enrich for primary transcripts that have also retained their original TTS. Indeed, as the first step of most in-vivo RNA degradation pathways is thought to consist of the removal of the 5' triphosphate (PMID: 19239894), the capturing of triphosphorylated RNA removes degraded and/or processed transcripts on the 3' end, particularly ends generated from RNase E processing. Nonetheless additional nucleases that do not require the removal of the 5 triphosphate have been shown to exist in E.coli (PMID: 29669055) and thus, some remaining 3'ends in our dataset maybe derived from processing*".

3. The authors use two programs to predict intrinsic terminators (Fig. 5E). I'm not familiar with these programs, but if they only rely on predicted structure, and not on structure followed by a U-rich sequence, then the discussion of these data should be softened. Finding a structured 3' end is expected of both true TTSs (at least for Rho-independent termination) and processed 3' ends (see Dar and Sorek, 2018 for more details on this for Rho-terminated RNAs), and hence should not be used as an indication that most TTSs are genuine. Lastly, a structured 3' end without a U-tract should not be referred to as a "termination structure" (pg. 4). If the programs look for a structure followed by a U-rich sequence, "termination structure" is appropriate.

The reviewer is right, we search for stem-loop structure (and not structure followed by a U-rich sequence), thus we agree with the reviewer to modify the sentence accordingly (see underlines) :
"Accordingly, 95% of the SMRT-Cappable-seq termination sites have a predicted stem-loop structure characteristic of termination sites (See below and Supplementary Fig. 3b)."

We have also soften the conclusion (see underlines) : "Taken together, these results suggest that most of the termination sites identified by SMRT-Cappable-seq are genuine TTS."

4. The authors have not done enough to convince me that increases in transcriptional context (TC) for translation-associated genes in rich media are not due to expression increases (pg. 6, para. 3). The graph presented in the point-by-point response clearly shows that genes with higher TC in rich medium are more highly expressed than in minimal medium. For example, I count 49 genes with a TC value in rich medium that is at least 2 higher than the TC value in minimal medium, and only 3 of these 49 genes are expressed less in rich medium than in minimal medium. As an aside, I was confused by this graph since it appears to show genes with TC differences that are non-integer values. I recommend removing the third paragraph of pg. 6 that describes this phenomenon.

We agree with the reviewer that we cannot rule out the confounding effect of drastic gene expression changes and redistribution of reads to a smaller subset of genes (Even though very unlikely this scenario is very unlikely*). We nonetheless felt that this observation is interesting (as pointed out by another reviewer). We therefore kept the figure but added a sentence to the manuscript to alert the reader of possible confounding effect :

"This result suggest that the operon complexity can depend on the growth condition. Nonetheless further work needs to be done to rule out the possible confounding effect of changes in gene expression."

*Total SMRT-Cappable-seq reads mapping to the E.coli genome were about the same for Rich (249,772 reads) and M9 (279,298 reads) growth conditions. Yet, the number of genes that have more contexts in Rich condition is 216 while the number of genes that have more context in M9 condition is 143. This data strongly suggest that Rich growth condition leads to a transcriptome with globally higher TC compare to M9 growth condition.

5. The authors have kept Fig. 5C, which shows transcripts on opposite strands terminating at the same site. I strongly disagree that this is an interesting finding. While I am a little confused as to whether the termination site itself is shared, as opposed to the terminator hairpin (I assume it's the former, but the point-by-point response refers to a sequence "between the sense and antisense"), neither of these would be unexpected. Bidirectional terminators are widespread, and lead to termination at different sites, but use the same hairpin. Terminating at the same nucleotide position but on opposite strands is not the same as TSS-dependent differences in read-through. The TSSs may be different, but so is the rest of the transcript! There is no reason to expect the level of read-through to be the same for the sense

and antisense transcripts. It's fine to use this as an example of an excludon, but it is not relevant for TSS-dependent read-through effects.

As requested by the reviewer, Fig. 5C is now used as an example of excludon and moved to Fig. 2d. Accordingly, we do not use anymore the concept of TSS-dependent read-through in the context of antisense transcripts as requested by the reviewer.

6. Pg. 8, first full paragraph. As I mentioned in my previous reviews, you would expect to find an enrichment for predicted attenuators in the TTS list because the TTSs are enriched for hairpins. The authors cannot tell whether TTS read-through is connected to predicted attenuator structure, so this paragraph should be removed.

We have used the PASIFIC algorithm to search for attenuator structures that are composed of a distinctive helix structure that holds the aptamer and functions as a switch between open and close state (see Millman et al. PMID: 27574119 and PMID: 27120414). These helix structures are not the same as the hairpins existing in the TTS (PMID: 27574119).

We now added an example of this helix structure in Supplementary Figure 7b to distinguish it from the simple hairpin structure.

In addition, we deleted one of the sequence predicted to be attenuator in the bacterial genome and measured the level of read-through in both Rich and M9 conditions using qPCR: We found that deleting the attenuator sequence significantly ($p.value < 0.005$) increased the read-through in Rich medium (Supplementary Figure 7c and d). This result indicates that this attenuator controls the conditional-regulated termination. In conclusion we have one example of TTS read-through level been connected to predicted attenuator structure. We cannot test all the predicted attenuators and therefore conclude accordingly “..suggesting that the read-through is modulated via such structures”. We have nonetheless soften further the conclusion “..suggesting that the read-through maybe modulated via such structures”.

Additional comments:

1. For Fig. S6, I still think it would be helpful to include the predicted structure(s). Note that there would be two structures if the authors' attenuation model is correct.

We have added the predicted Terminator and Attenuator structure using the attenuation model (PMID: 27574119) in the Supplementary Figure 7. And there are two structures to control the switch between open and close state of the transcription.

2. The authors still refer to riboswitches inappropriately. At the top of pg. 8 they describe a program they used to find attenuators, not riboswitches as they suggest. For the paragraph at the bottom of pg. 7/top of pg. 8, there is no need to focus on riboswitches. It would be more appropriate to describe

attenuators. Not all riboswitches regulate transcription, and not all attenuators are associated with riboswitches.

We have removed from the manuscript the use of the term 'riboswitch' as suggested.

3. Pg. 2, second full paragraph. The authors should mention here that transcripts ending in A will have their TTSs miscalled. This does not impact any of the conclusions, but is important for readers to know, especially if they plan to use the TTS list for downstream analyses.

We have added the following sentence : "Because SMRT-Cappable-seq library preparation procedure involved the non-template addition of A at the 3' end of the RNA, the exact position of the read start can be off by one or several nucleotides in the case of a TTS finishing at A."

4. Top of pg. 5, "found 1335 (56%) new operons from which 454 are extended by SMRT-Cappable-seq". I was confused by this – what is the difference between finding a new operon and extending a newly found operon?

New operons include extending operons as well as the operons having new internal TSS or TTS. We have described the comparison of operons in the Supplementary Notes 'SMRT-Cappable-seq operons' part.

For clarity, we change this sentence to the following sentence : "We also compared the SMRT-Cappable-seq operons with operons predicted in a previous study {Conway, 2014 #415} , and we found around 40% of our operons were not predicted (see Fig. 2b for example)".

Reviewer #2 (Remarks to the Author):

The authors have constructively addressed all my concerns and I believe this paper is suitable for publication in its present form. The only minor improvement I suggest is to replace ref 26 with PMID: 18158897. Ref 26 used single molecule approach to argue that 3 different intrinsic terminators they studied worked by different mechanisms (forward translocation, RNA shearing, and allosteric mechanism). This is highly unlikely and bulk of biochemical evidence argues that the only mechanism of intrinsic termination is the one described in PMID: 18158897. Conceptually similar "allosteric" mechanism also works for Rho-mediated termination (PMID: 20075920).

We thank Reviewer 2 for the suggestion, and we have changed the original ref 26 to PMID: 18158897 now.

Reviewer #3 (Remarks to the Author):

This manuscript presents data that complement and extend the existing information. The ability to map the end of the transcript that starts kilobases upstream is unique and cannot be overstated. It is likely that the majority of the determined transcript ends correspond to primary termination sites. The complexity of operon structures that was largely overlooked because of the short sequencing reads is quite striking.

The manuscript has been significantly improved, the results will be of interest to many research groups, and are easy to follow after the revisions. I have a few minor points for consideration:

5' UTRs: it is actually shocking that, given the size selection, the authors successfully identified any early TTS. Perhaps it would be helpful to state more directly that 5' UTR are typically very short, below the selected 1 kb size.

We have modified the sentence to clearly state that the size of the premature transcripts are below the 1kb threshold: *“The relatively low abundance of premature termination sites in SMRT-Cappable-seq libraries is contrasting with the large number of known regulatory sites controlling transcript elongation in the 5'UTR of genes. This inconsistency can be explained by the size selection step during library preparation depleting prematurely terminated transcripts that are typically very short, below the selected 1 kb threshold”*.

Factors affecting TSS-dependent readthrough: I thought this came up before, but this could have been a while. In addition to the strength of the promoter, which I am not sure will contribute that strongly because few promoters are strong enough (cooperation was shown for rrn promoters), the initial transcribed sequence (obviously different in each case) has been shown to have a direct effect on termination (PMID: 2467004). One could look whether transcribed regions that increase readthrough have a potential to base pair with the terminator stem; basically, very distant antiterminators.

Agreed. This is an interesting suggestion that we are envisioning to test in the future. We have added the reference suggested by the reviewer to the discussion together with the following paragraph : *“It is likely that in native context, the read-through is under the control of a number of additional factors such as the strength of the promoter that determines the degree of cooperation between RNA polymerase molecules in transcription elongation (PMID: 2467004)”*

5' nucleotide: it may be more appropriate to cite recent Belasco papers (e.g. Luciano 2017) instead of Deana 2008 as they have information. Although this is not critical, mentioning alternative 5' caps could ass interest.

As suggested, we have now cited Luciano and Belasco's paper (PMID: 28673541) instead.

Termination: Santangelo review is outdated; perhaps cite PMID 27023849 instead, or primary references.

As suggested, we have now added the more recent review (PMID: 27023849) for the termination.